# A retinoid analogue, TTNPB, promotes clonal expansion of human pluripotent stem cells by upregulating CLDN2 and HoxA1

Suman C. Nath[1,2], Shahnaz Babaei-Abraki[1], Guoliang Meng[1], Kali A. Heale[1], Charlie Y. M. Hsu[1] & Derrick E. Rancourt [1,2] ✉

Enzymatic dissociation of human pluripotent stem cells (hPSCs) into single cells during routine passage leads to massive cell death. Although the Rho-associated protein kinase inhibitor, Y-27632 can enhance hPSC survival and proliferation at high seeding density, dissociated single cells undergo apoptosis at clonal density. This presents a major hurdle when deriving genetically modified hPSC lines since transfection and genome editing efficiencies are not satisfactory. As a result, colonies tend to contain heterogeneous mixtures of both modified and unmodified cells, making it difficult to isolate the desired clone buried within the colony. In this study, we report improved clonal expansion of hPSCs using a retinoic acid analogue, TTNPB. When combined with Y-27632, TTNPB synergistically increased hPSC cloning efficiency by more than 2 orders of magnitude (0.2% to 20%), whereas TTNPB itself increased more than double cell number expansion compared to Y-27632. Furthermore, TTNPB-treated cells showed two times higher aggregate formation and cell proliferation compared to Y-27632 in suspension culture. TTNPB-treated cells displayed a normal karyotype, pluripotency and were able to stochastically differentiate into all three germ layers both in vitro and in vivo. TTNBP acts, in part, by promoting cellular adhesion and self-renewal through the upregulation of *Claudin 2* and *HoxA1*. By promoting clonal expansion, TTNPB provides a new approach for isolating and expanding pure hPSCs for future cell therapy applications.

Human pluripotent stem cells (hPSCs) including human embryonic stem cells (hESCs)[1], and human induced pluripotent stem cells (hiPSCs)[2] are attractive tools for regenerative therapy, disease modeling and drug discovery[3–5]. When first discovered, hPSCs were passaged and cryopreserved as cell clumps because hPSCs undergo anoikis, a massive cell death when cells are dissociated[1]. This hindered many downstream applications. The discovery of Rho kinase (ROCK) inhibitor, Y-27632, partially solved this problem by preventing anoikis following single cell dissociation[6]. Although its mechanism of action is not well understood, Y-27632 appears to: (1) stabilize E-cadherin when cells are enzymatically dissociated[7], and (2) relax the actin cytoskeleton thus avoiding cellular blebbing[6,8]. Consistent with this mechanism, our previous observations indicate that cell-cell contact, not anti-apoptosis, is predominantly responsible for the Y-27632's promotion of hPSC growth in both static and suspension culture[9].

Although Y-27632 permits hPSC survival upon single cell dissociation, it promotes multi-cell colony formation, not single-cell colony formation. "Multi-cell colony" refers to a group of cells in a colony derived from multiple individual cells of the same origin. In such a colony, the cell population is heterogeneous. By contrast a "single-cell colony", or "clonal colony" is a group of cells in a colony that descend from and are genetically identical to a single progenitor cell. In the presence of Y-27632, about 70% of hPSCs participate in colony formation at passaging cell density ($1–5 \times 10^4$ cells/cm$^2$)[10], which is a considerably high number of cells. However, cloning efficiency is often very low when cells are plated at clonal density (50–100 cells/cm$^2$) in the presence of Y-27632. Inability to clone hPSCs

[1]Department of Biochemistry and Molecular Biology, Cumming School of Medicine, University of Calgary, Calgary, Canada. [2]McCaig Institute for Bone and Joint Health, University of Calgary, Calgary, Canada. ✉e-mail: rancourt@ucalgary.ca

 1

presents a major problem when deriving genetically modified hPSC lines since transfection and genome editing efficiencies are not satisfactory with Y-27632. As a result, colonies tend to contain heterogeneous mixtures of both modified and unmodified cells, making it difficult to isolate the desired clone buried within the colony. Ideally, each colony should arise through the clonal expansion of a single progenitor cell. However, this requires seeding cells at low density, resulting in an extremely low colony-forming efficiency due to the lack of cell-cell contact. The generation of clonal colonies would greatly facilitate the purification and analysis of genetically modified cells.

Here, we report enhanced clonal colony expansion from dissociated single hPSCs using the retinoid analogue TTNPB. The retinoids work through either retinoic acid receptors (RAR) or retinoid-X receptors (RXR). TTNPB selectively activates RAR-type receptor which is specific for either 9-cis- or all trans-retinoic acid[11]. TTNPB was first used in chemical reprogramming of mouse (m) iPSCs[12], which increased reprogramming efficiency significantly. Here, we tested the hypothesis of preventing apoptosis of dissociated single hPSCs using TTNPB. We observed clonal expansion and enhanced aggregation of hPSCs in TTNPB-treated cells. TTNPB acts, in part, by promoting cellular adhesion and self-renewal through the upregulation of *Claudin 2* and *HoxA1*.

## Results

### TTNPB promotes higher clonal colony formation compared to Y-27632

We have previously observed that Y-27632 promotes hPSC adhesion and cell-cell contact prior to colony formation[8]. When passaged at a routine density ($3-5 \times 10^4$ cells/cm²), over 70% of ESCs (H9) or iPSCs (4YF) participate in colony formation (Fig. 1a, b). However, at clonal density (50–100 cells/cm²), cell survivability was very low (0–0.3%). When cells are grown at a clonal density, no or only very few cell colonies were observed several days after cell seeding (Fig. 1a, b). We observed that the retinoic acid analogue, TTNPB, significantly promotes the clonal colony forming efficiency of hPSCs (Fig. 1c, d). We first optimized the concentration of TTNPB for clonal expansion of hESCs and hiPSCs. Human ESCs showed an increase of colony forming efficiency at 0.5 μM compared to other concentrations tested (Fig. 1c). However, for hiPSCs, there were differences observed between 0.25 and 0.5 μM and each of the other concentrations: 1 and 2 μM (Fig. 1d). Overall, hiPSCs showed high clonal colony-forming efficiency at 0.5 μM.

When applied alone, TTNPB improved clonal colony formation compared to Y-27632 (2–3% vs. 0–0.3%) (Fig. 1e, f). However, when applied in combination, clonal colony formation increases significantly from 2–3% to 15–17% in both hESCs and hiPSCs. Clonal expansion of hPSCs using TTNPB and Y-27632 required a long exposure time. When TTNPB was removed along with Y-27632 on day 1 or day 2, the colony-forming efficiency decreased for both ESCs and iPSCs (Fig. 1g, h). The effective synergy between Y-27632 and TTNPB occurred over 4 days, as the number of clonal colonies increased steadily over time.

To determine whether clonal colony appeared from a single cell or not, we cultured a mixture (1:1) of GFP labeled cells and unlabeled cells onto Laminin-511-coated plates at a density of 50–100 cells/cm² in mTeSR1 medium supplemented Y-27632 and TTNPB (Fig. 1i). Clonal colony formation was monitored daily thereafter up to day 4, which is when GFP fluorescence dropped below a detectable range. On day 1, we observed single cells that survived and expressed GFP under fluorescence microscopy. We tracked single cells from day 2 to 4 and observed that the attached single cells grew to small colonies. We observed homogeneous colonies containing either all GFP labeled cells, or all unlabeled cells (Fig. 1j). No colony with a heterogeneous mixture of labeled and unlabeled cells was observed.

The interaction of cell-ECM is also an important factor for clonal expansion of dissociated single hPSCs. To explore this hypothesis, we cultured dissociated single hPSCs in both Matrigel and Laminin-511 with TTNPB and Y-27632. We observed that hPSCs showed higher survival and clonal colony formation in Laminin-coated plates compared to Matrigel in both TTNPB and Y-27632-treated conditions (Supplementary Fig. 1).

### TTNPB promotes improved expansion of hPSC in both adherent and suspension culture

Since dissociated single hPSCs are prone to apoptosis, Y-27632 can rescue single cells from apoptosis during routine passage. However, the cell survival ratio of hESC is greatly improved with TTNPB 24 h after seeding compared to Y-27632 (Fig. 2a). The combination of Y-27632 and TTNPB also showed a higher survival ratio when compared to the Y-27632 itself. hiPSCs also showed a similar trend in cell survival either after treatment with TTNPB or TTNPB + Y-27632 following 24 h of seeding single cells (Fig. 2b). We performed a live/dead assay to confirm the reason behind the low expansion of hPSCs with Y-27632. We observed that most of the single hPSCs showed higher apoptotic signals 24 h after plating in Y-27632-treated conditions, whereas TTNPB showed lower apoptotic and higher live cell signals (Supplementary Fig. 2a). The combination of Y-27632 and TTNPB also showed lower apoptotic signals compared to Y-27632 alone. This observation was in line with our results from Western blot for cell survival showing the reduction of cleaved caspase-3 in the presence of TTNPB in comparison to Y-27632-treated cells for both ESCs (Supplementary Fig. 2b, d) and iPSCs (Supplementary Fig. 2c, e). We also observed reduced expression of cleaved caspase-3 in the combination of Y-27632 and TTNPB in both ESCs and iPSCs. To test if TTNPB increases adhesion of single hPSCs, we performed adhesion assay using crystal violet in a 96-well plate. We observed higher cell adhesion in TTNPB-treated cells 6 h after seeding compared to Y-27632 for both hESCs and iPSCs (Supplementary Fig. 2f, g). The combination of Y-27632 and TTNPB also showed higher adhesion of hPSCs 24 h after seeding.

TTNPB-treated hESCs expanded 2x higher than Y-27632 after 4 days of seeding at passaging density (Fig. 2c). The combination of Y-27632 and TTNPB also showed a relatively higher final cell number, similar in magnitude to TTNPB. hiPSCs also showed 1.5x higher final cell number in TTNPB-treated conditions compared to Y-27632 (Fig. 2d). The combination of Y-27632 + TTNPB also showed a higher final cell number than Y-27632. TTNPB-treated hESCs showed compact and round shape colonies, whereas Y-27632- and Y-27632 + TTNPB-treated cells showed loose and irregular shape colonies after 24 h of seeding (Fig. 2e). Y-27632-treated hESCs showed a widespread F-actin cytoskeletal network compared to TTNPB-treated colonies, which showed a compact F-actin network surrounding the cells (Fig. 2f). Y-27632-treated cells also showed widespread myosin throughout the cell cytoplasm, whereas, TTNPB-treated cells showed contractility/less expression of myosin in the cell cytoplasm.

Like adherent culture, TTNPB improved survival and expansion of hPSC in suspension culture (Fig. 3). We first optimized the TTNPB concentration for efficient growth of hPSC in static suspension culture. Like adherent culture, we found better aggregate formation and expansion of both hESCs and hiPSCs at 0.5 μM compared to other concentrations (Supplementary Fig. 3). Lesser than 0.5 μM TTNPB, we observed smaller and less numbers of aggregates in both hESC and hiPSC lines. Similar phenomena were observed with higher than 0.5 μM TTNPB in both cell lines. TTNPB-treated cells showed about 80% survival rate 24 h after seeding single cells in suspension culture, whereas Y-27632 showed about 60% survival rate for both hESCs and hiPSCs (Fig. 3a, b). After culturing for 5 days in suspension culture, we observed greater than 2-fold higher final cell numbers with TTNPB compared to Y-27632 (Fig. 3c, d). The combination of Y-27632 + TTNPB showed a slightly higher final cell number for both cell lines. Interestingly, hPSC aggregate morphology differed between Y-27632 and TTNPB treatment as we observed smoothen edges with empty cores in both Y-27632 and Y-27632 + TTNPB-treated conditions, whereas TTNPB-treated aggregates showed uneven edges with cell-filled cores (Fig. 3e).

### Characterization of TTNPB-expanded clonal colonies

Clonally expanded hPSCs expressed pluripotency markers which were confirmed by immunostaining of Oct4, Nanog, TRA-1-60, and SSEA-4 after culturing for 7 days (Fig. 4a). Flow cytometry analysis of hPSCs also indicated high level expression (>95%) of pluripotency markers: Oct4,

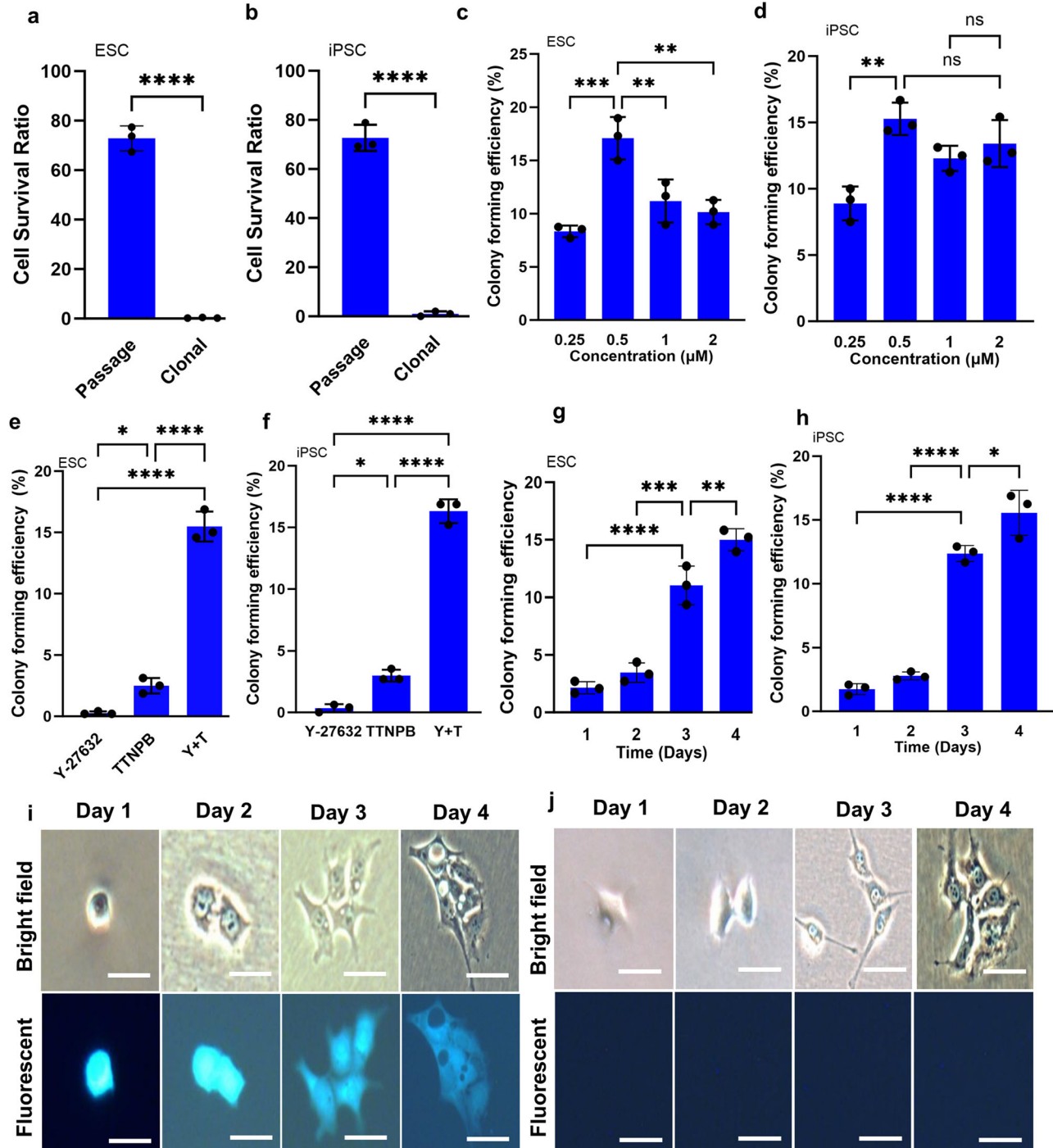

**Fig. 1 | Clonal colony formation using Y-27632 and TTNPB. a, b** Colony formation at high cell density and at clonal cell density in the presence of Y-27632 after 96 h of culture. **c, d** Effect of different concentrations of TTNPB on clonal colony formation of ESCs and iPSCs. **e, f** Clonal colony formation using TTNPB (0.5 μM) and Y-27632 (10 μM) at clonal density. **g, h** Effect of treatment duration of TTNPB on clonal colony formation at 0.5 μM. **i, j** Cell tracking using GFP-labeled cells in TTNPB-treated conditions. Colonies are either all GFP labeled, or all unlabeled. No colony with a heterogeneous mixture of labeled and unlabeled cells was observed. Scale bars: 100 μm. Data represented from $N = 3$ experiments. $*P < 0.05$, $**P < 0.01$, $***P < 0.001$ and $****P < 0.0001$ were considered as significant. N. S. Not significant.

Nanog, SSEA-4, and TRA-1-60 (Fig. 4b). We performed karyotyping for the TTNPB-treated cells and the results indicated that the clonally expanded cells showed normal karyotypes (Fig. 4c). We also checked the differentiation capability of both cell lines in vitro and in vivo. Embryoid bodies (EBs) generated from the cells of both cell lines differentiated spontaneously into the three germ layers, AFP (ectoderm), smooth muscle actin (mesoderm), and β-III tubulin (endoderm) (Fig. 4d). We also used the teratoma assay as a complementary in vivo test. The produced teratomas also contained cells from all three germ layers (Fig. 4e).

## TTNPB promotes clonal colony expansion by upregulating *Claudin-2* and *HoxA1*

To investigate the mechanism of action of TTNPB-mediated enhanced clonal colony expansion, we performed RNA-sequencing. TTNPB-treated

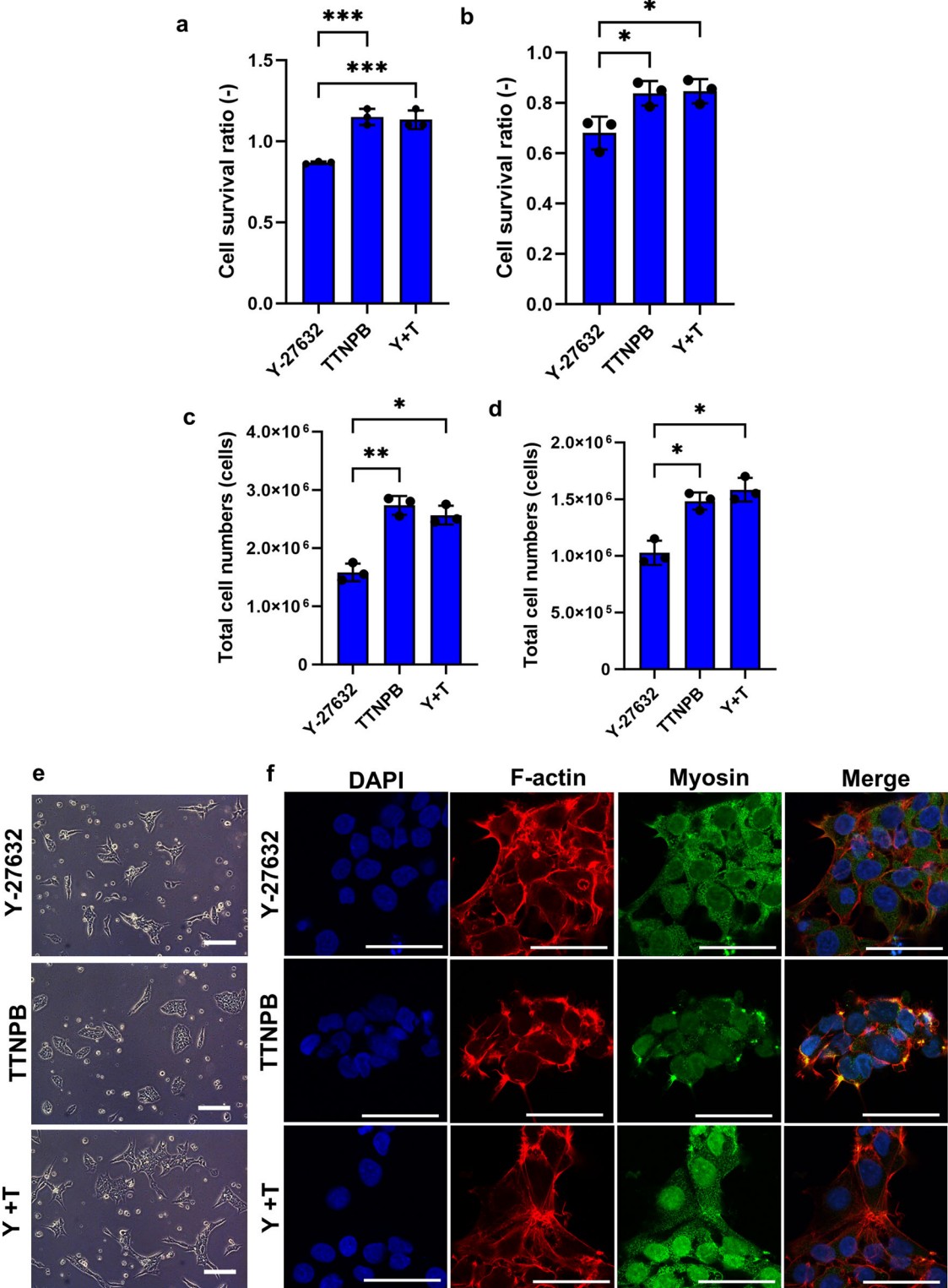

**Fig. 2 | Single-cell passaging with Y-27632 and TTNPB in adherent culture at passaging density. a**, **b** Cell survival ratio of dissociated single cells cultured with Y-27632, TTNPB and Y-27632 + TTNPB 24 h after seeding. **c**, **d** Final cell number after culturing with Y-27632, TTNPB and Y-27632 + TTNPB after 96 h. **e** Bright field images of ESCs 24 h after seeding with Y-27632, TTNPB and Y-27632 + TTNPB.

Scale bars: 200 μm. **f** Immunostaining of ESCs for F-actin and Myosin after treating with Y-27632 and TTNPB 24 h after seeding. Scale bars: 50 μm. Data represented from *N* = 3 experiments. *$P < 0.05$, **$P < 0.01$, and ***$P < 0.001$ were considered as significant. N. S. Not significant.

cells showed about 6-fold upregulation of claudin-2 (*CLDN2*) and Homeobox protein A1 (*HoxA1*) compared to untreated conditions (Fig. 5a). Other highly upregulated genes were: (i) nuclear transcription factors: *RAR-beta, STAT1, TCF7, HOXA2, HOXD1, CTNNB1, ZNF503*; (ii) cytoplasmic:

*CYP26A1/C1, CPE, PAK, TSPAN1, ATP6V0A4*; (iii) plasma membrane: *ADGRD1, HAS2*; and (iv) extracellular: *WNT11, FNDC5, FGF4, LEFTY5, COL1A1* (Fig. 5b). The most important canonical pathways upregulated in TTNPB-treated hESCs were related to WNT-β catenin, pluripotency, Rho

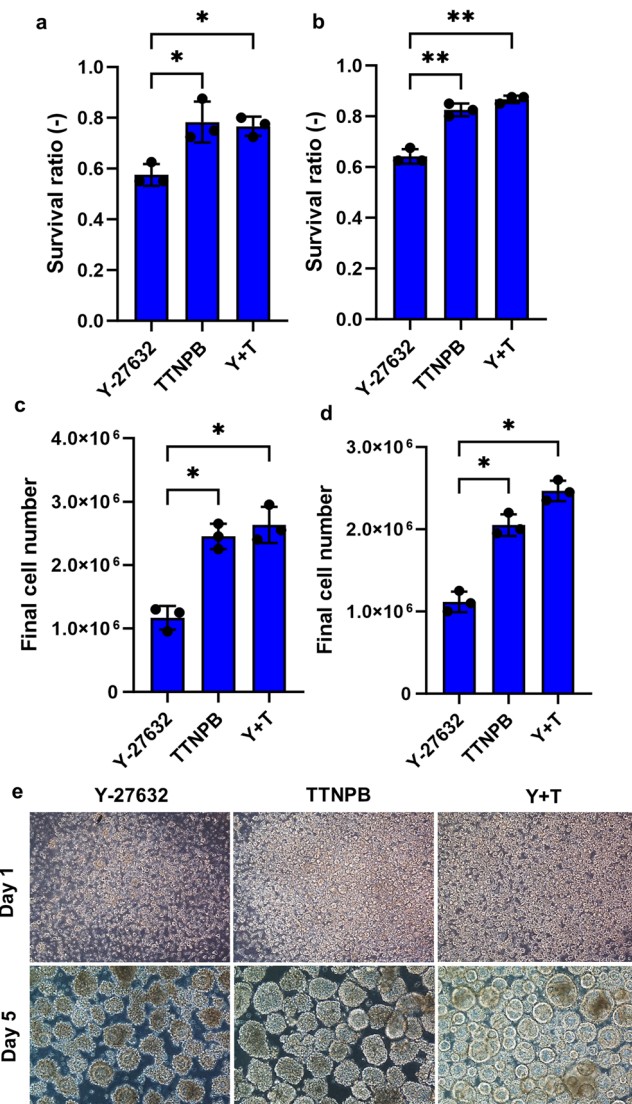

**Fig. 3 | Single-cell passaging with Y-27632 and TTNPB in suspension culture. a, b** Survival ratio of dissociated single cells cultured with Y-27632, TTNPB and Y-27632 + TTNPB 24 h after seeding. **c, d** Final cell number of hPSCs after culturing with Y-27632, TTNPB and Y-27632 + TTNPB after 120 h. **e** Morphology of ESC aggregates on 24 h and 120 h of seeding. Scale bars: 200 μm. Data represented from N = 3 experiments. *P < 0.05 and **P < 0.01 were considered as significant. N. S. Not significant.

family GTPase, and Integrin signaling (Supplementary Fig. 4a), which promoted cytoskeletal balance and better adhesion through integrin-ECM interaction (Supplementary Fig. 4c). Genes upregulated with TTNPB treatment mostly contributed to preventing cell death and promoting cell migration, which increased cell proliferation capability (Supplementary Fig. 4b).

We confirmed the upregulation of *CLDN2* and *HoxA1* using RT-PCR. We observed higher expression of *HoxA1*, *CLDN2*, and *Bcl-2* in TTNPB-treated conditions compared to Y-27632 (Supplementary Fig. 5a).

We then disrupted *HoxA1* and *CLDN2* genes in hESCs using CRISPR/Cas9 to investigate their roles of TTNPB-mediated cell proliferation (Figs. 6, 7). PCR surveyor assays showed successful disruption of *HoxA1* and *CLDN2* genes in two clones among three that we isolated (Supplementary Fig 5c, d). We confirmed the *HoxA1* gene deletion in the *HoxA1* knockout (HKO) clones by DNA sequencing (Fig. 6a, b). We observed that 8 nucleotides (TCAAGTTG) were deleted after the

nucleotide "C" in the HKO clones compared to the wild type (WT). We also confirmed successful *HoxA1* gene deletion in other HKO clone.

The HKO cells showed poor adhesion and proliferation in TTNPB-treated adherent culture conditions compared to Y-27632 (Fig. 6c, e). Like adherent culture, suspension HKO cells also showed poor aggregation on day 1, and poor yield on day 3 in TTNPB-treated condition compared to Y-27632 (Fig. 6d, f).

Like *HoxA1*, we also succeeded in disrupting the *CLDN2* gene using CRISPR/Cas9 (Fig. 7). We confirmed the *CLDN2* gene deletion in the *CLDN2* knockout (CKO) clones by DNA sequencing (Fig. 7a, b). We observed that 4 nucleotides (CACT) were deleted after the nucleotide "C" in the CKO clones compared to the WT. We confirmed successful *CLDN2* gene deletion from the other CKO clone. The CKO cells showed poor adhesion and proliferation in TTNPB-treated adherent culture conditions compared to Y-27632 (Fig. 7c, e). We also observed no aggregation and proliferation of CKO cells in TTNPB-treated condition compared to Y-27632 in suspension culture (Fig. 7d, f).

We observed higher expressions of *E-cadherin*, *Bcl−2*, *YAP*, *Ki-67* and *cyclin D1* in TTNPB-treated cells compared to Y−27632 (Fig. 8a). The expression of these genes was downregulated in both HKO and CKO cell lines in the presence of TTNPB (Fig. 8b, c). We also observed higher expression of *YAP* in the nucleus of TTNPB-treated cells compared to Y-27632 (Supplementary Fig. 5b). Overall, we observed high expression of *CLDN2* and *HoxA1* in TTNPB-treated cells which, in turn, upregulated tight and adherens junction proteins that promoted high survival and expansion of hPSCs (Fig. 8d).

## Discussion

Dissociated single hPSCs are prone to apoptosis as cells lose cell-ECM connection and cell surface ligands that are necessary for cell-cell contact after routine passage using cell dissociative enzymes. The Rho-associated protein kinase inhibitor, Y-27632, has been shown to efficiently enhance the survival of hPSCs after single cell dissociation in passaging density. Y-27632 appears to stabilize E-cadherin when cells are enzymatically dissociated, and hPSCs are protected through Rho-ROCK signaling mediated cell-ECM interaction and cell-cell interaction[7]. Our previous[9,10] and recent findings[8] are consistent with this mechanism. However, Y-27632 is inefficient in promoting clonal expansion when cells are seeded at low density, which we overcome using TTNPB in this study.

At routine passaging density ($1 \times 10^4$ cells/cm$^2$), over 70% of hPSCs participate in colony formation, consisting of four steps: adhesion of individual cells through cell-ECM interaction, migration of adjacent cells near each other, cell aggregation via cell-cell contact, and proliferation (Fig. 2). However, at the clonal density (50–100 cells/cm$^2$) needed to promote clonal expansion, colony-forming efficiency is very low (0–0.3%) under Y-27632-treated conditions (Fig. 1). This phenomenon is also visible for TTNPB-treated cells, although TTNPB increased the survival ratio a little more. This means that it is difficult for individual adherent cells to form colonies from a single cell state due to the lack of cell-cell contact. This result suggests that cell-cell contact is essential for hPSCs survival and expansion.

When cells are seeded at clonal density in the presence of TTNPB, the clonal colony-forming efficiency is enhanced greatly (Fig. 1) by simulating self-renewal in isolated single cells. Once a cell division has occurred, TTNPB is no longer required to promote clonal colony formation. When combined with Y-27632, TTNPB improved the clonal expansion of hPSCs by more than two orders of magnitude. Y-27632 and TTNPB have synergistic effects on the enhancement of clonal colony formation. Our results indicate that Y-27632 improves the attachment of single cells at clonal density but not cell proliferation. Rather than improving the attachment of cells, TTNPB compliments Y-27632 in clonal colony formation by stimulating the proliferation of attached single cells.

TTNPB is an analog of retinoic acid and a RAR agonist[11]. Previously, TTNPB has been shown to enhance chemical reprogramming of miPSCs[12]. Based on our observations in this study, we hypothesize that TTNPB may

**Fig. 4 | Characterization of clonally expanded hPSCs using TTNPB. a** Immunostaining of clonally expanded colonies of hESCs for the pluripotent markers, Oct4, Nanog, SSEA-4 and TRA-1-60. **b** FACS analysis of hESCs for the expression of pluripotency markers Oct4, Nanog, SSEA-4 and Tra-1-60. **c** Karyotyping of clonally expanded hESC colonies treated with TTNPB. **d** In vitro differentiation of hESC EB staining for β-III tubulin (ectoderm; i), AFP (endoderm; ii), smooth muscle actin (mesoderm; iii). **e** In vivo differentiation via teratoma formation: pigmented cells (ectoderm, i), cartilage (mesoderm; ii) and columnar glands (endoderm, iii). Scale bars: 100 μm. Data represented from *N* = 3 experiments.

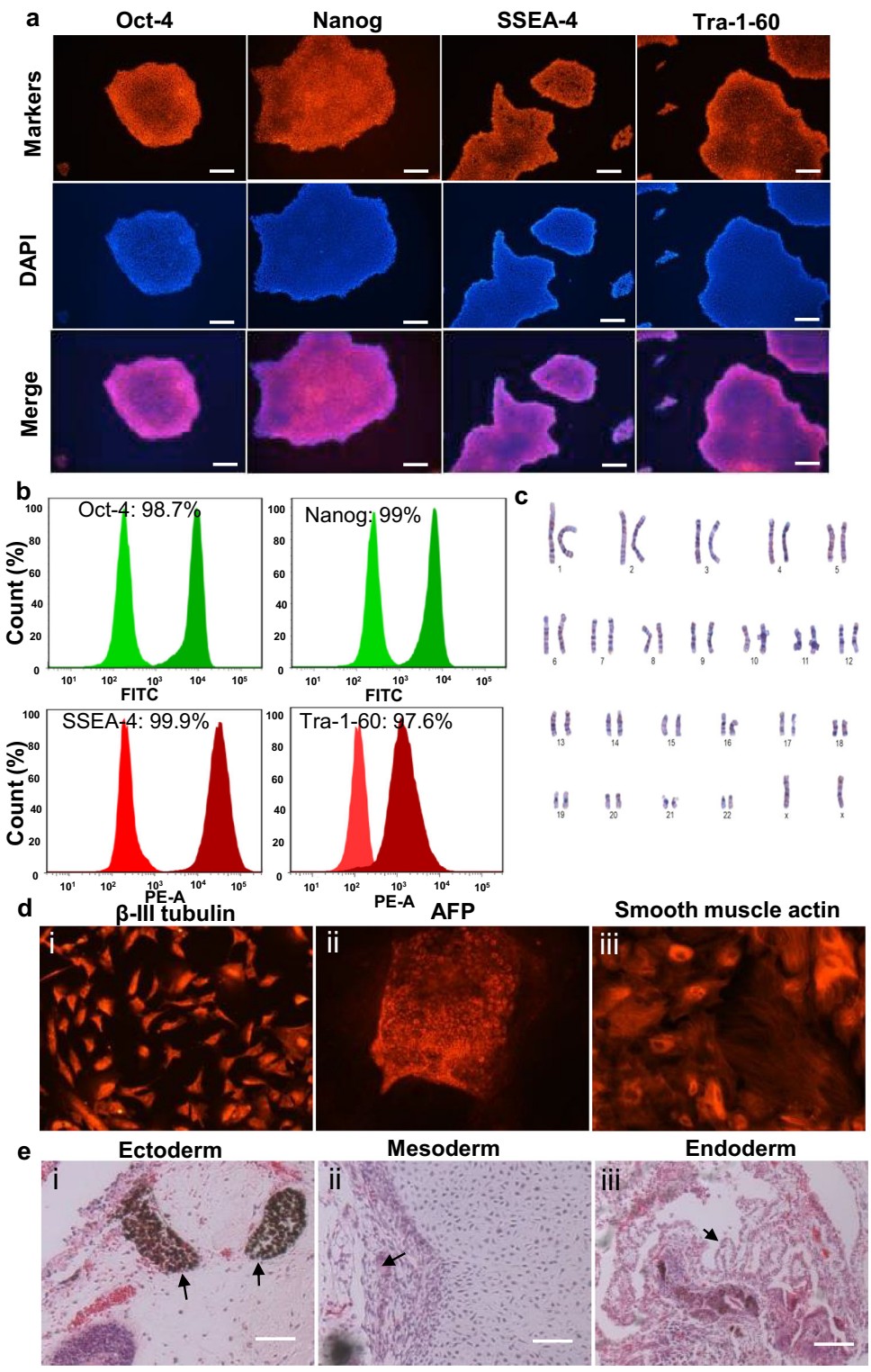

act by increasing the clonal expansion of reprogrammed single progenitor cells into individual colonies. TTNPB has no similarity with pioglitazone, a peroxisome proliferator-activated receptor gamma (PPARγ) agonist, which was reported to enhance the cloning efficiency of hPSCs, along with Y-27632[13]. In their report, a density of $6 \times 10^4$ cells/well of a 6-well plate was employed to test the cloning efficiency in the presence of pioglitazone and Y-27632. However, the cell density they adopted is close to the passaging cell density and is around 100 times the clonal density that we used in this study. This means that not every colony they obtained is derived from a single progenitor cell.

Cell tracing experiments were performed to demonstrate that the colonies do form from a single progenitor cell in the presence of Y-27632 and TTNPB. Dissociated single cells seeded at clonal density in the presence of Y-27632 and TTNPB not only adhered to the Laminin surface, but proliferated and formed single cell-derived colonies, which contain either all GFP labeled cells, or all unlabeled cells. No colonies with a heterogeneous mixture of labeled and unlabeled cells were observed. This suggests that TTNPB can be efficiently used for deriving pure iPSCs from a trace amount of patient cells for clinical application. Possibly acting through YAP upregulation, TTNPB promotes higher hPSC self-renewal allowing proliferation

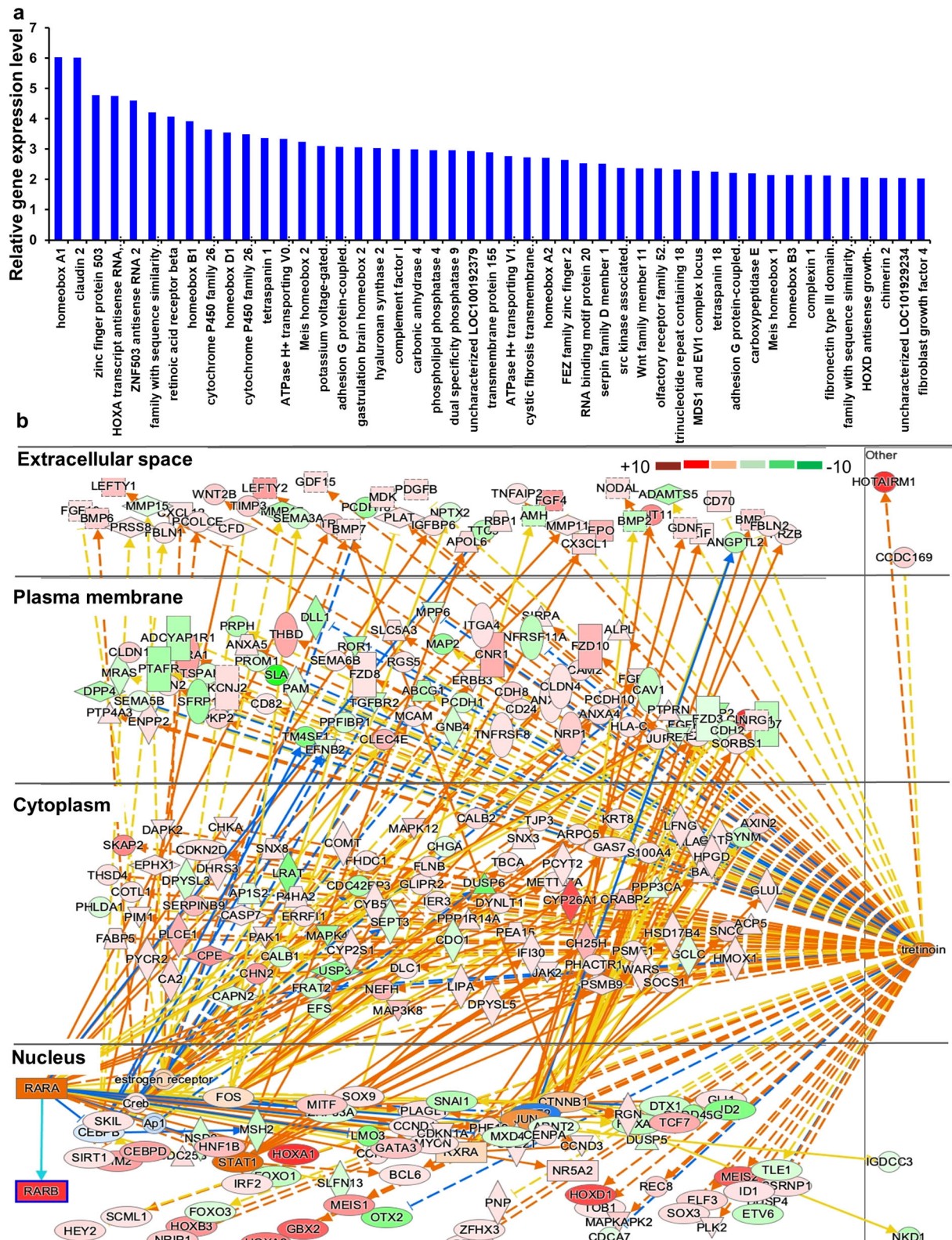

**Fig. 5 | RNA-seq analysis data from TTNPB-treated hPSCs. a** Relative gene expression fold in TTNPB-treated condition compared to without treatment. **b** Graphical representation of upregulated genes at different parts of the cell in TTNPB-treated condition.

at the single cells stage. In turn, we observed lower apoptotic signals in TTNPB-treated conditions compared to Y-27632 (Supplementary Fig. 2).

Since every colony appears from a single cell, TTNPB also offers the potential of obtaining high numbers of hPSCs in both adherent and suspension cultures (Figs. 2, 3). We observed higher survival of hPSCs 24 h after seeding single cells, and more than two times higher final number of cells in TTNPB-treated condition compared to Y-27632 in both adherent and suspension culture. The high survival ratio in TTNPB-treated condition lessened the lag phase of dissociated single cells 24 h after seeding in both culture systems. We have previously noted that Y-27632-treated aggregates

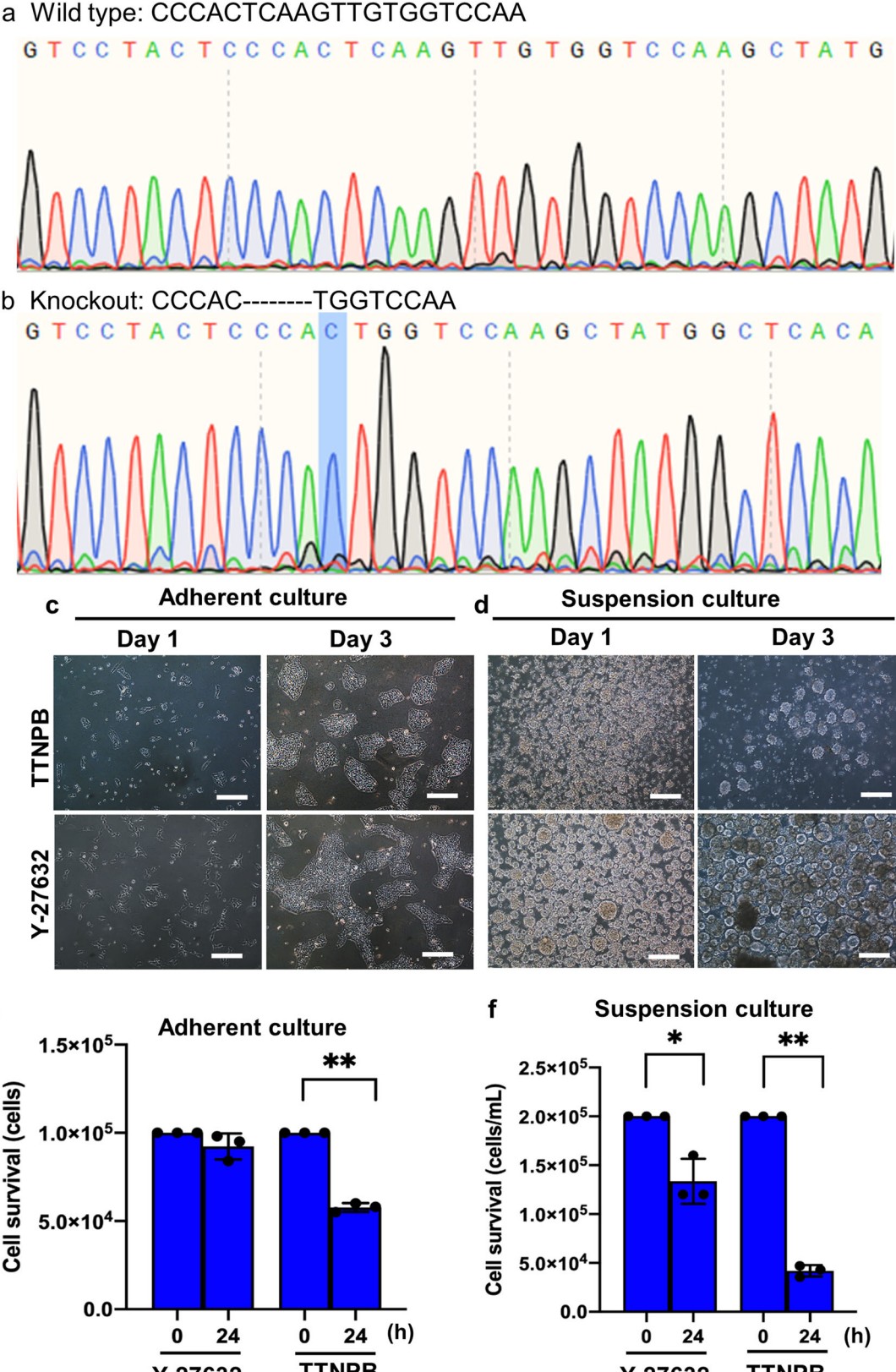

**Fig. 6 | Disruption of *HoxA1* gene using CRISPR/Cas9 in hESC line.**
**a, b** Chromatogram of DNA-seq data obtained from wild type and *HoxA1* knockout clones. Highlighted nucleotide "C" shows the deletion start point for 8 nucleotides in the HoxA1 knockout clones. **c, d** Brightfield images of *HoxA1* knockout cells (HKO) after treating with TTNPB and Y-27632 on day 1 and day 3 in adherent and suspension culture. **e, f** Cell survival data 24 h after seeding single cells of HKO in adherent and suspension culture. Scale bars: 100 μm. Data represented from *N* = 3 experiments. *\*P* < 0.05 and *\*\*P* < 0.01. N. S. Not significant.

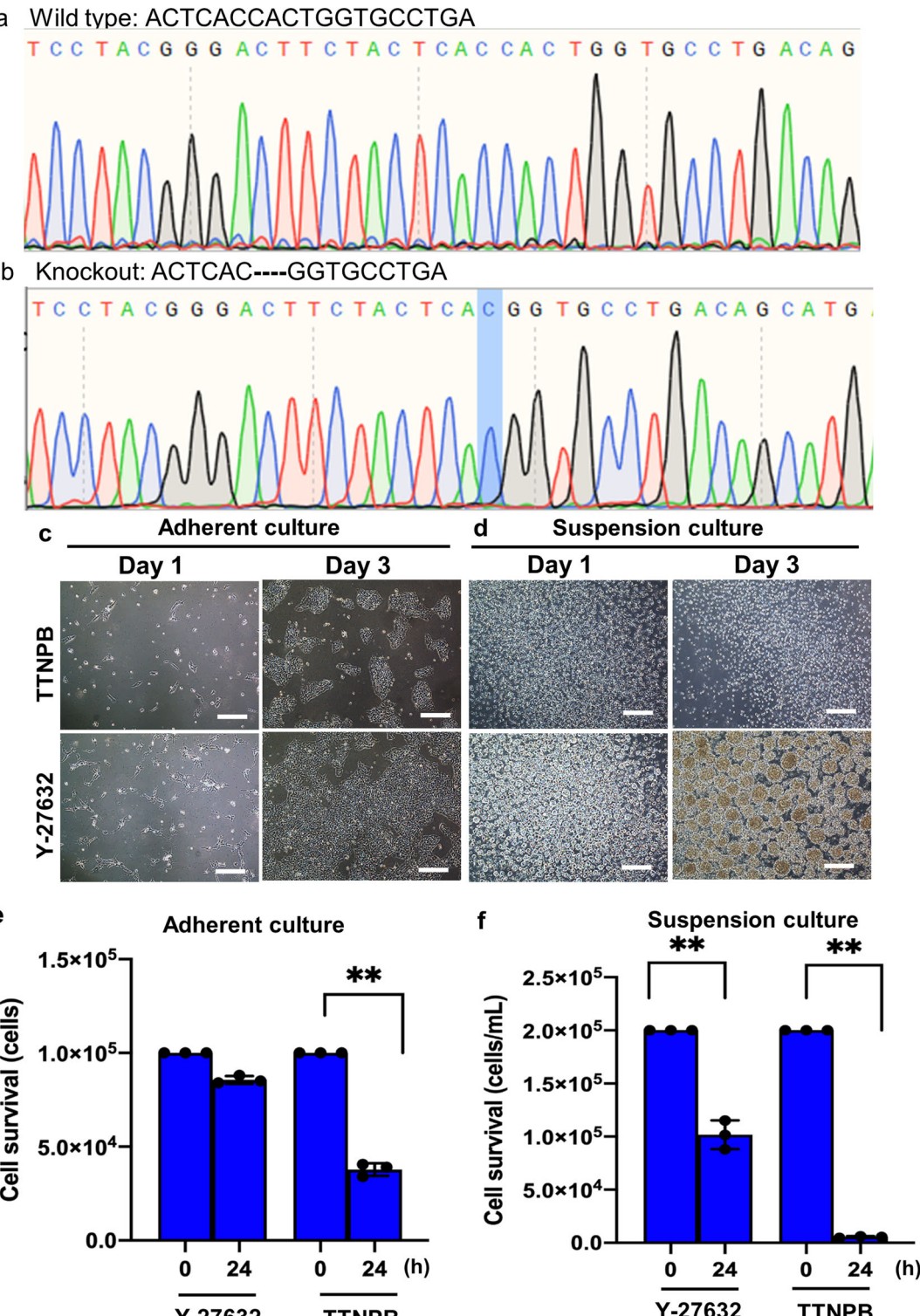

**Fig. 7 | Disruption of *CLDN2* gene using CRISPR/Cas9 in hESC line.**
**a**, **b** Chromatogram of DNA-seq data obtained from wild type and *CLDN2* knockout clones. Highlighted nucleotide "C" shows the deletion start point for 4 nucleotides in the CLDN2 knockout clones. **c**, **d** Brightfield images of *CLDN2* knockout cells (CKO) after treating with TTNPB and Y-27632 on day 1 and day 3 in adherent and suspension culture. **e**, **f** Cell survival data 24 h after seeding single cells of CKO in adherent and suspension culture. Scale bars: 100 μm. Data represented from $N = 3$ experiments. $**p < 0.01$.

possess a thick extracellular matrix (ECM) on the periphery preventing nutrient exchange and resulting in empty cores[14]. However, TTNPB-treated aggregates present roughened edges, and their core is filled with cells which is indicative of sufficient nutrient exchange. Since nutrient diffusion depends on aggregate size and accumulation of ECM over the suspension culture time[15], TTNPB may be superior to Y-27632 for providing the high

cell numbers by less secretion of ECM. Ai *et al.* also observed core-filled aggregates while investigating their new medium with Activin/IWP2/CHIR (AIC) compared to conventional medium (StemFlex and E8) for culturing hPSCs in suspension[16]. They reported that cells cultured with the conventional media showing empty core underwent lumenogenesis which they confirmed by staining with the polarity markers, *ZO-1* and *PODXL1*. As we

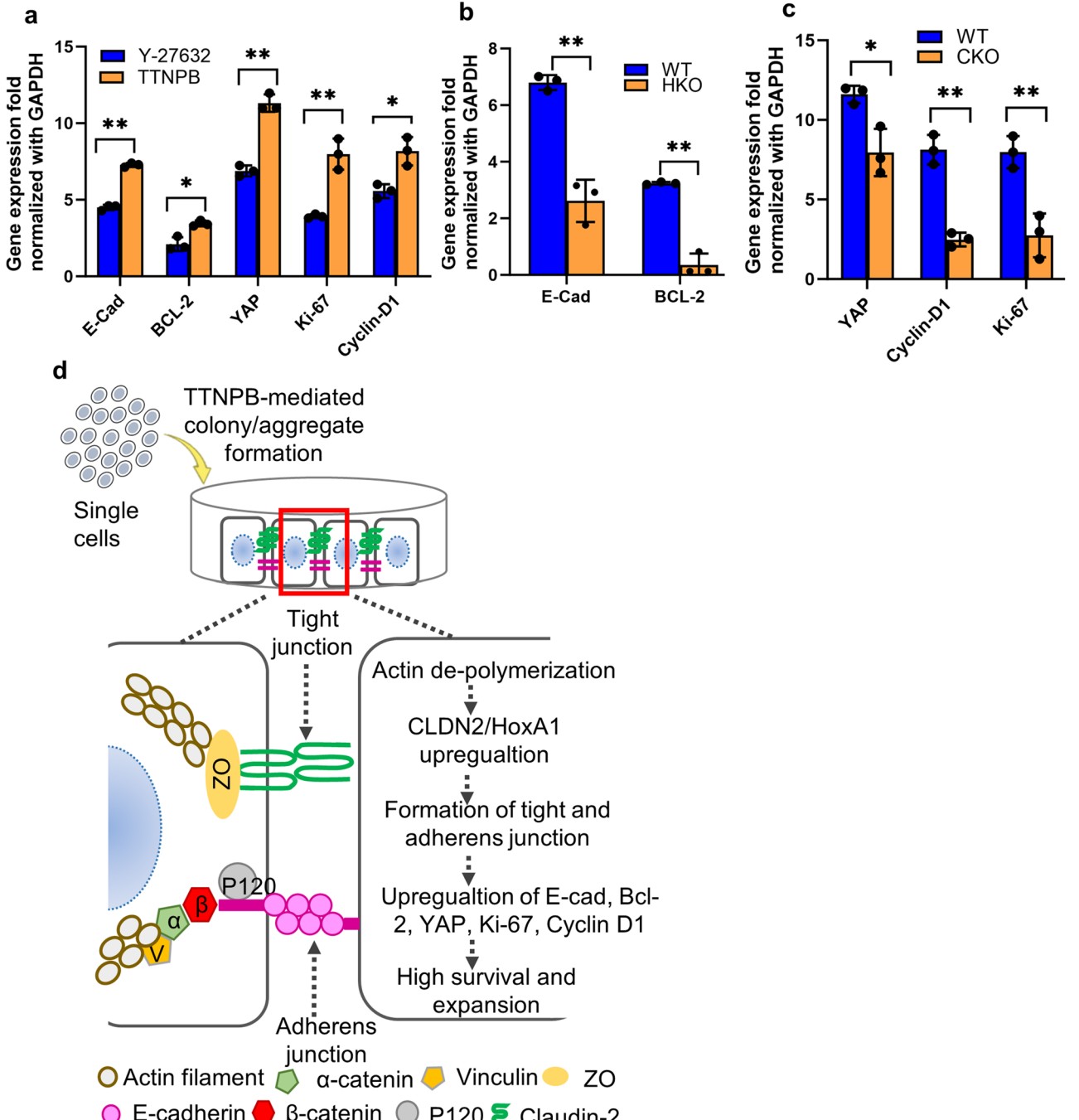

**Fig. 8 | Mechanism of TTNPB-mediated cell survival. a** Relative gene expression of *E-cadherin, Bcl-2, YAP, Ki-67* and *Cyclin-D1* in Y-27632- and TTNPB-treated cells detected by RT-qPCR. **b** Relative gene expression of *E-cadherin* and *Bcl-2* in wild type and HKO cells in TTNPB-treated condition. **c** Relative gene expression of *YAP,* *Cyclin D1* and *Ki-67* in wild type and CKO cells in TTNPB-treated condition. **d** Schematic illustration of TTNPB-mediated survival of hPSC. Scale bars: 100 μm. Data represented from $N = 3$ experiments. *$P < 0.05$, and **$P < 0.01$ were considered as significant. N. S. Not significant.

observed core-filled aggregates in TTNPB-treated aggregate, we assume that TTNPB might promote less lumenogenesis. Since TTNPB does not negatively affect pluripotency or karyotype stability, it may provide a better platform for obtaining highly pure hPSCs for cell therapy applications (Fig. 4).

While seeking the reasons behind TTNPB-mediated clonal expansion of hPSCs, we found two interesting candidate genes by RNA-seq analysis: *CLDN2* and *HoxA1* (Fig. 5). *CLDN2* is one of several tight junction (TJ) proteins expressed in specific epithelial cells and plays important roles in the paracellular barrier controlling the flow of molecules in the intercellular space between cells[17]. Importantly, claudins promote aggregate formation

and self-renewal in cancer stem cells[18,19]. This self-renewal is mediated by elements of the Hippo pathway, namely *YAP* (Yes-associated protein), and its paralog, *TAZ* (Tafazzin), (referred to as *YAP/TAZ*), transcriptional co-factors which translocate from TJs to the nucleus associating with *TEAD* or other transcription factors[20]. In PSCs, *YAP/TAZ* regulate self-renewal (*Cyclin D1, Ki-67*) and pluripotency (*Oct4, Sox2, Nanog*), thereby promoting long-term expansion[21–24]. Like these, we observed higher expression of *YAP, Cyclin D1* and *Ki-67* in TTNPB-treated cells. We also observed nuclear translocation of *YAP* in the TTNPB-treated condition as evidenced by higher expression of *YAP* in the nucleus compared to Y-27632 (Supplementary Fig. 5b). Consistent with this, we observed lower expression of

*YAP, Cyclin D1* and *Ki-67* in the TTNPB-treated CKO cells compared to wild type cells.

*CLDN2* has also recently been reported to suppress Rho signaling, via *ZO-2* resulting in dampened cytoskeleton dynamics and promotion of epithelial cyto-architecture[25,26]. We also observed similar changes in cytoskeletal dynamics (myosin) in the TTNPB-treated condition (Fig. 2f). Although both Y-27632 and TTNPB prevent PSC apoptosis by controlling myosin activity, TTNPB dampens the myosin activity while Y-27632 controls cellular contraction by relaxing the myosin, which prevents cellular blebbing in turn. While TTNPB upregulates *CLDN2* to prevent apoptosis, Y-27632 has a different mechanism of preventing apoptosis, not related to *CLDN2* upregulation. With the upregulation of *CLDN2*, the promotion of tight and adherens junctions increase in the TTNPB-treated cells, which is also a major difference with Y-27632 in preventing hPSC apoptosis. Taken together, we hypothesize that overexpression of *CLDN2* by TTNPB promotes hPSCs self-renewal by controlling cytoskeletal dynamics, and translocating *YAP* into the nucleus.

Not been previously studied in hPSCs, *HoxA1*, a member of the homeobox transcription factor family, is involved in mammalian embryogenesis and anterior pattern formation[27]. In many cancers, upregulation of *HoxA1* has been associated with the suppression of apoptosis via upregulation of the anti-apoptotic genes *Bcl2* and *BAX*, and the promotion of proliferation via upregulation of *Cyclin D1*[28,29]. We also observed upregulation of *Bcl2* in hPSCs treated with TTNPB (Fig. 8). *HoxA1* also upregulates E-cadherin expression in breast cancer cells[30], and we observed similar upregulated expression of E-cadherin in the TTNPB-treated condition. This may provide another survival mechanism for dissociated single hPSCs treated with TTNPB. By considering these, we hypothesize that TTNPB upregulates HoxA1, which in turn upregulates E-cadherin and *Bcl2*, promoting cellular adhesion and survival. The expression of E-cadherin and *Bcl-2* was suppressed significantly when the *HoxA1* gene was disrupted using CRISPR. We also observed significant loss of cell growth in the HKO cells.

In conclusion, TTNPB enhances clonal expansion of hPSCs for clinical application. It works by upregulating *CLDN2* and *HoxA1*, which in turn prevent apoptosis by controlling cytoskeletal dynamics, promoting cellular adhesion, and activating self-renewal. Further investigation will reveal the signaling pathways involved in TTNPB-mediated clonal expansion, which for now promises to be a useful culture additive for expanding hPSCs.

## Methods

### Culture and maintenance of hPSCs

The hESC line H9 was obtained from WiCell Research Institute, USA and the hiPSC lines 4YF was obtained from Dr. James Ellis' laboratory at the University of Toronto (Toronto, Canada). Both hESCs and hiPSCs were routinely cultured and maintained in chemically defined, feeder-free Laminin-511™ (Nacalai) with mTeSR™1 medium (STEMCELL Technologies) under standard culture conditions (37 °C, 5% $CO_2$). We coated 60 mm culture dishes with Laminin-511™ prior to cell culture for at least 2 h at room temperature. Cells were grown on ECM-coated dishes containing complete mTeSR™1 medium. Cells were passaged on every 4–5 days as single cells via enzymatic treatment. Briefly, the spent medium was aspirated and cells were washed once with DPBS -/- (Gibco), and then treated with TrypLE Express (Gibco) with 0.5 mM EDTA (Invitrogen) and 10 μM ROCK inhibitor, Y-27632 (STEMCELL Technologies). After incubating the cells for 5–7 min at 37 °C, mTeSR1 was added and cells were detached from the plate by pipetting. After centrifugation, cells were resuspended in fresh mTeSR™1 medium and counted by automatic cell counter (Thermofisher), and seeded onto Laminin-511-coated plates at $1 \times 10^4$ cells/cm$^2$ in mTeSR™1 medium. On the next day, medium was replaced with fresh medium without Y-27632 and maintained until the next passage.

### Cell seeding at clonal density

Human PSCs were dissociated into single cells, and then seeded onto 35 mm culture dishes coated either with Laminin-511 or Matrigel. Cells were seeded at a density of 50–100 cells/cm$^2$ after counting with an automatic cell counter for the clonal expansion. Two mL of mTeSR™1 medium was added with 10 μM ROCK inhibitor and 0.5 μM TTNPB (Tocris Biosciences). Two to three days after seeding, the spent medium was replaced with fresh mTeSR™1 medium without the two chemicals. Then the media were changed every other day thereafter. Colonies were counted on day 7. Then colony forming efficiency was calculated by dividing the total number of colonies with the total number of seeding cells.

### Seeding GFP-labeled cells mixed with unlabeled cells

To determine whether hPSC colonies were clonally expanded from a single progenitor, hESCs were labeled by transient transfection via nucleofection (Amaxa) using the reporter plasmids gWiz-GFP (Genlantis), which expresses the green fluorescent protein (GFP) reporter. Following nucleofection, cells were re-plated at a density of $5 \times 10^5$ cells/cm$^2$. One day after nucleofection, cells were dissociated into single-cell suspension and FACS sorted (Sony) to enrich GFP-positive cells to 98% purity. Afterwards, transfected GFP$^+$ cells were mixed with non-transfected unlabeled cells at a ratio of 1:1, then seeded onto 35 mm dishes at a density of 50 GFP-labeled and 50 unlabeled cells/cm$^2$. Colony formation was monitored daily thereafter up to day 4, when GFP fluorescence level dropped below the detectable range. Brightfield and GFP expressing images were captured every day to detect the clonal expansion of hPSC using a confocal microscopy (Olympus).

### Adherent and suspension culture of hPSCs

Human PSCs were dissociated into single cells using TrypLE Express with 0.5 mM EDTA and 10 μM Y-27632. Then cells were counted using an automatic cell counter. For the expansion of cells in adherent culture with passaging density, single cells at a density of $1 \times 10^4$ cells/cm$^2$ were seeded with mTeSR™1 medium in a 6-well plate coated with Laminin™ with Y-27632, TTNPB or Y-27632 + TTNPB, and then incubated at standard culture conditions (37 °C, 5% $CO_2$). The spent medium was changed every day from day 2 without the two chemicals and cultured until day 4. Cells were dissociated into single cells using TrypLE Express with 0.5 mM EDTA and 10 μM Y-27632 on day 1 to determine adhesion ratio. Cells were counted using an automatic cell counter. Cell adhesion/survival ratio was calculated by dividing total cell number on day 1 with the total number of seeded cells on day 0. Final number of cells was determined by counting the total number of cells on day 4.

For the optimizing the different concentrations of TTNPB, single cells ($2 \times 10^5$ cells/ml) were seeded in ultra-low attachment 6-well plate (Corning) with 0.25, 0.5, 0.63, and 0.75 μM of TTNPB and cultured for 5 days. The morphology of the aggregates were observed everyday under a brightfield microscope (Olympus). For the expansion of cells in suspension culture, single cells at a density of $2 \times 10^5$ cells/ml were seeded with mTeSR1 medium in an ultra-low attachment 6-well plate with Y-27632, TTNPB or Y-27632 + TTNPB, and then incubated at standard culture conditions (37 °C, 5% $CO_2$). Two days after seeding, the spent medium was replaced with fresh mTeSR1 medium without the two chemicals. Then the media was changed every day thereafter. Aggregates were dissociated into single cells using TrypLE Express with 0.5 mM EDTA and 10 μM Y-27632 on day 1 to determine the survival ratio. Cells were counted using an automatic cell counter. Cell survival ratio was calculated by dividing total cell number on day 1 with the total number of seeded cells on day 0. Final number of cells was determined by counting the total number of cells on day 5.

### Immunofluorescent staining

Cell colonies were fixed with 4% paraformaldehyde (PFA) for 15 min at room temperature, washed three times with 1x phosphate buffered saline (PBS), permeabilized with 0.1% Triton-X 100 (Sigma Aldrich) for 15 min at room temperature, and then washed three times with PBS. Cells were blocked with 10% fetal bovine serum (FBS) for 30 min at room temperature

to minimize non-specific binding of antibodies. Fixed hPSC colonies were incubated with primary antibodies against Oct4 (Milipore), Nanog (Milipore), SSEA-4 (Milipore), TRA-1-60 (Milipore), and YAP (Cell Signaling) at 1:100 dilution and kept at 4 °C overnight. Differentiated embryoid bodies (EBs) were incubated with primary antibodies (all from Sigma-Aldrich) against ®-tubulin III (ectoderm marker), smooth muscle actin (mesoderm marker), and α-fetoprotein (endoderm marker) at 1:400 dilution and kept at 4 °C overnight. The next day, cells were washed and incubated with the secondary antibodies, Alexa Fluor 488, 546 and 594 (1:200; Invitrogen) for at least one hour at room temperature. Cells were then washed 3 times with PBS before adding DAPI in Slowfade. For monitoring F-actin and myosin, cells were stained with Phalloidin (Invitrogen) and anti-non-muscle Myosin IIA antibody (Abcam), respectively. Then the cells were incubated with Slowfade containing DAPI before imaging with confocal microscopy (Nikon).

## Flow cytometry
Pluripotency markers, Oct4, Nanog, SSEA4 and Tra-1-60, were analyzed by fluorescence-activated cell sorting (FACS). Briefly, cell colonies were dissociated into single cells using TrypLE Express, were fixed with 4% PFA, washed three times with PBS, and permeabilized with 0.6% Saponin (Sigma Aldrich). After three washes with PBS, cells were then resuspended in PBS containing 3% BSA for 30 min at 37 °C. Resuspended cells were incubated with the following antibodies (Millipore) for at least 1 h at 37 °C: anti-Oct4 (Alexa Fluor 488 conjugate; 1:50 dilution, Millipore), anti-Nanog (FITC conjugate; 1:50 dilution, Millipore), anti-SSEA-4 (PE conjugate, 1:50 dilution, STEMCELL Technologies) and anti-Tra-1-60 (PE conjugate, 1:50 dilution, STEMCELL Technologies). Mouse IgG Alexa Fluor 488, mouse IgG1 FITC, mouse IgG3PE and mouse IgM PE, kappa were used as isotype controls for the primary antibodies. Flow cytometric analysis was performed using FACSAria III (BD Biosciences).

## In vitro differentiation
Cell colonies were dissociated and plated onto 35 mm agar-coated dishes (low attachment) in a differentiation medium consisting of DMEM supplemented with 15% fetal bovine serum (FBS), 1mM L-GlutaMax, 0.1 mM 2-mercaptoethanol, and 1 mM nonessential amino acids (all reagents were from Life Technologies). Five to seven days later, when some of the aggregates formed cystic embryoid bodies (EBs), they were collected, re-plated onto gelatin-treated four-well plates in the same medium and cultured for 6–10 days. Then the EBs were fixed, and immunocytochemical staining was performed on early differentiation markers of the three germ layers as stated above.

## In vivo differentiation
In vivo differentiation was performed by teratoma assay. Cell colonies were dissociated into single cells, then collected by centrifugation and injected into the rear leg muscles of 6- to 8-week-old SCID-beige mice ($2 \times 10^6$ cells per injection). Teratomas were removed from injection sites after 10–12 weeks, fixed overnight in 4% PFA. The samples were then embedded in paraffin, sectioned, and examined histologically after staining with Eosin and Haematoxylin for the three germ layers.

## RNA sequencing
Confluent ESCs were dissociated into single cells using TrypLE Express and seeded in a 6-well plate in passaging density with or without TTNPB as described above. On the next day, RNA extraction was performed after treatment with Trizol (Sigma). At least triplicate samples were harvested for preparing RNAs. Then RNA sequencing libraries were prepared from purified RNA using NEBNext Ultra™ Directional RNA Library Prep Kit (New England Biolabs) for Illumina sequencing according to the manufacturer's instructions. The individual libraries were sequenced by NextSeq 500 (Illumina). Base-scaling and demultiplexing were done with bcl2fastq v.2.17.1.14 software (Illumina). The RNA-seq data were analyzed by Ingenuity Pathway Analysis software (Qiagen).

## Generation of *CLDN2* and *HoxA1* knockout cells, DNA sequencing and PCR Surveyor assay
The *CLDN2* and *HoxA1* knockout cell lines were developed using CRISPR/Cas9 technology. Three human *CLDN2* and *HoxA1* short gRNA targets on Exon 1 were designed, synthesized, ligated into pX458 vector-pSpCas9(BB)-2A-GFP (Addgene) and confirmed by DNA sequencing. Oligo synthesis and DNA sequencing were done at the DNA Laboratory in the University of Calgary. The gRNA sequences for *CLDN2* are: (i) CTAGGATGTAGCCCACAAGT (ii) GGTGCTATAGATGTC ACACT and (iii) ACGGGACTTCTACTCACCAC. The gRNA sequences for *HoxA1* are: (i) GCTTGGACCACAACTTGAGT (ii) CATTCAC-CACTCATATGGAC, and (iii) GAGTCGCCACTGCTAAGTAT. Then the CRISPR activity of both gRNAs were tested. From 3 gRNAs tested, gRNAs (ii) and (iii) were selected for transfection of H9 cells for *CLDN2* knockout. For the *HoxA1* knockout, gRNAs (i) and (ii) were used for generation of *HoxA1* knockout. After transfection of the CRISPR constructs, GFP positive cells were sorted into 96-well plates by Flow Cytometry. Two knockout clones for both *CLDN2* and *HoxA1* were isolated and examined. Cells were expanded, and DNA sequencing was performed thereafter for confirming the disruption of *CLDN2* and *HoxA1*.

For PCR Surveyor assay, DNA fragments around short guide DNA sites were amplified by PCR. G and C DNA fragments were also amplified as positive control. All DNA samples were cleaved with Surveyor nuclease according to Manufacturer's recommendations (Integrated DNA Technologies) and then analyzed on 10% TBE acrylamide gel.

For DNA sequencing, DNA fragments around the short gRNA site of *CLDN2* and *HoxA1* knockout clones were amplified by PCR. Then the PCR products were subcloned into pEGFP-C2 vector (Clonetech). After that, the plasmid DNA from individual clones were sent for Sanger DNA sequencing at the DNA laboratory, University of Calgary to confirm all indels. The DNA sequencing data were analyzed by Chromatography software.

## Culture of *CLDN2* and *HoxA1* knockout cells
The *CLDN2* and *HoxA1* knockout cells were cultured on Laminin-511™-coated 6-well plates with mTeSR™1 medium. When confluent, cells were dissociated into single cells and then counted using an automatic cell counter. Then single cells at a density of $1 \times 10^4$ cells/cm² were seeded with mTeSR™1 medium in a 6-well plate coated with Laminin™ with either Y-27632 or TTNPB, and then incubated at standard culture conditions. Cells were observed under a brightfield microscope and counted on day 1 and 3 after dissociating into single cells.

## Cell adhesion assay
Cell adhesion assays were done using a protocol described elsewhere[31]. In brief, hPSCs were treated with TTNPB (0.25-0.5 µM) in the presence or absence of Y-27632. Twenty-four hours after, cells were dissociated using TrypLE Express, collected and counted. A total of $5 \times 10^4$ cells were seeded into each well of a 96-well plate coated with the Laminin and incubated for 6 h. Then, cells were washed two times with DMEM/F12 to remove non-adherent cells. Adherent cells were fixed with 10% formalin for 15 min, and post-fixed in 100% ethanol for 5 min at room temperature. Following fixation, cells were then stained with 0.4% crystal violet in methanol for 5 min at room temperature. Cells were then rinsed with demineralized water followed by solubilizing by adding 1% SDS, and the optical density at 570 nm was measured by a microplate reader (Bio-Rad).

## Western blot
For analysis of western blots, the cells were lysed with Radio Immuno-precipitation Assay (RIPA) Buffer. The lysates were sonicated twice for eight seconds and centrifuged to remove any remaining cell pellet. The Halt™ Protease Inhibitor Cocktail (Thermo Fisher Scientific) was added to each sample (1:1000) to prevent protein degradation. Bradford's method was employed to assess total protein content. 30 µg of total protein from each sample was separated using 12% SDS-PAGE followed by electro-transferred onto a nitrocellulose membrane. The membrane

was incubated with blocking buffer containing 5% skim milk powder overnight at 4 °C. Then, the membranes were probed with specific antibodies against cleaved-caspase3 (1:500; Invitrogen) and β-actin (1:200 dilution; Santa Cruz) for 1.5 h at room temperature. Finally, the membrane was incubated with specific secondary antibodies for 1 h. After washing with TBST buffer three times, quantification of the results was performed by a densitometric scan of films. For analysis we employed Image J software, measuring the integrated density of bands after background subtraction.

## Quantitative real time PCR
RT-qPCR was used to determine the relative gene expression of *E-cadherin*, *Bcl-2*, *YAP*, *Ki-67* and *Cyclin-D1* in Y-27632- and TTNPB-treated cells. First, total RNA was extracted using the Single Cell RNA Purification Kit (Norgen Biotek Corp.) according to the manufacturer's instructions. RNA concentrations were then determined by spectrophotometer. The RNA was transcribed into cDNA using the protoscript II c-DNA synthesis kit (New England Biolabs). cDNA generation consisted of the following steps: 65 °C for 5 min, 42 °C for 60 min, and 80 °C for 5 min. Finally, fast SYBR™ Green Master Mix (Thermo Fisher Scientific) was used for the RT-qPCR gene expression analysis (performed on the Applied Biosystems™ StepOnePlus™ Real-Time PCR System). The mRNA expression levels were all quantified relative to the housekeeping gene, GAPDH.

## RNA isolation and RT-PCR
hESCs were cultured on a plate with Y-27632 and TTNPB. Then single cells were collected for RNA isolation on day 3. Total RNA was extracted using RNeasy Mini kit (Qiagen) according to the manufacturer's instructions with DNase I digestion. RNA was measured using a NanoPhotometer P-Class (IMPLEN), and then 500 ng RNA were used for cDNA synthesis using Superscript® IV Reverse Transcriptase (Thermo Fisher Scientific) and Oligo(dT)20 Primer (50 μM) (Thermo Fisher Scientific) according to the manufacturers' instructions. GAPDH was used as a housekeeping gene. Primer sequences used in the amplification reactions are *HOXA1* (sense) GGGAAAGTTGGA-GAGTACGGC; *HOXA1* (antisense) CCTCAGTGGGAGGTAGTCAG; *CLDN2* (sense) TCCGTCTGTCTGTCTGTGTG; *CLDN2* (antisense) ATTGATCACCTTGTGTGTGC; GAPDH (sense) ACAGTCAGCCGCATCTTCTT; GAPDH (anti-sense) GACAAGCTTCCCGTTCTCAG. The PCR products were separated on 1% agarose, stained with ethidium bromide, and then visualized and photographed on a UV transilluminator.

## Statistics and reproducibility
All experiments were performed at least triplicate and data represented as mean ± SD. Exact *N* values for each experiment are defined and described in the figure legends. We used Student's *T* test for evaluating the significance from three replicate experiments. $P < 0.05$, $P < 0.001$ and $P < 0.0001$ were considered as significant.

## Reporting summary
Further information on research design is available in the Nature Portfolio Reporting Summary linked to this article.

## Data availability
The raw fastq files of RNA-seq data are available in the GEO under the accession number SUB11997390. The source data for the graphs and charts in the main figures can be found in Supplementary Data 1. The original gel pictures from both RT-PCR and Western blot can be found in Supplementary Figs. 6 and 7. The gating strategy for flow cytometry data can be found in the Supplementary Figs. 8–16.

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

## Acknowledgements
The authors are thankful to Shiying Liu and Yaping Yu from the University of Calgary Genome Engineering Center for doing the teratoma assay and CRISPR/Cas9 experiments, respectively. The authors are also thankful to Dr. Igor Kovalchuk's group at the University of Lethbridge for RNA-sequencing experiments. Suman Nath was partially supported by a Cumming School of Medicine-Postdoctoral Fellowship. This work is supported by the Canadian Institute of Health Research granted to D.E.R.

## Author contributions
Conceptualization and design: SCN, GM and DER; methodology: SCN, SBA, GM, KH, CH and DER; data curation: SCN, SBA, GM, KH and CH; writing- original draft preparation: SCN and GM; Original draft editing: SCN and DER; funding acquisition: DER.

## Competing interests
The authors declare no competing interests.

## Additional information

Ethical compliance The University of Calgary's Ethics Oversight Committee approved the ethics protocol (Human Ethics: REB14-1914) for culturing hPSCs in the Rancourt lab. Teratoma analysis was conducted under the Animal Care Protocol AC19-0134

