## [Peer Review File · Communications Biology]

Reviewers' comments:

Reviewer #1 (Remarks to the Author):

In this manuscript, the authors reported that TTNPB, a retinoic acid agonist, promotes colony expansion from dissociated human PSCs and is useful for clone isolation. They identified two factors, CLDN2 and HOXA2, as effector molecules for the TTNPB's action. The authors claimed that CLDN2 regulates cytoskeletal dynamics and YAP activity while HOXA1 upregulated the genes associated with cell-cell adhesion and proliferation. The effect of Y27632+TTNPB seems to be drastic and impressive, but the manuscript includes some concerns that require clarification.

1. The authors showed that combined treatment of dissociated hPSCs with Y27632+TTNPB showed a greater effect on colony formation than each reagent alone (Figure 1). However, the initial adhesion and cell survival/proliferation are not so different when cells treated with TTNPB only or Y+T were seeded (Figure 2, 3). What is the cause for the difference in colony formation between TTNPB alone and Y+T treatment?

2. ROCK inhibitors have been reported to block PSC death induced by myosin-dependent cellular contraction. A simple way to show that TTNPB works through different mechanisms from ROCK inhibitors is to determine whether TTNPB affects myosin activity or cellular contraction of dissociated PSCs.

3. KO experiments seem key for the author's claim about the molecular mechanism of TTNPB action, but the manuscript lacks important information on this part. The detailed descriptions of KO line generation are required (gRNA targets, how many KO clones are isolated and examined? and so on). It is also needed to confirm the expression of HOXA1/CLDN2 is actually defective at protein levels.

4. The authors proposed that TTNPB upregulates HOXA1/CLDN2, which in turn promotes the formation of TJ and AJ (shown in Figure 8D). However, no data are showing TJ/AJ formation in TTNPB-treated cells. As they mentioned in Introduction, ROCK inhibitors have been also shown to promote cell-cell adhesion. If the final output caused by these two reagents are similar (promoting/stabilizing cell-cell adhesion?), why TTNPB are more effective than ROCK inhibitors?

5. Suppl Figure 1 is too weak to claim that TTNPB diminished apoptosis of dissociated PSCs. It would be also confusing that green fluorescent signals were detected in the Y+T sample. More quantitative data for cell death/survival are required.

6. Figure legends are not sufficient and reader-friendly. Which timepoint the number of colonies was counted? The concentration of TTNPB used is not specified (except Figure 1C-D). What is 'survival ratio' and how did the authors determine it (Figure 3)? Materials and methods section is also insufficient. The procedures for the quantification of cell death or survival, KO clone generation, DEG identification from RNA-seq data and PCR surveyor assay should be described.

Reviewer #2 (Remarks to the Author):

In this study, Nath et al. reported improved clonal expansion and suspension culture of hPSCs using a retinoic acid analogue, TTNPB, which acts, in part, by promoting cellular adhesion and self-renewal through the upregulation of Claudin 2 and HoxA1. I have the following concerns about the data in the

manuscript

1. How is the colony forming efficiency (Fig. 1A, B) calculated ? It is difficult to calculate the colony forming efficiency when passaged at a high cell density ($3\sim 5 \times 10^4$ cells/cm²), I want to know how the colony forming efficiency (over 70%) is calculated in this study.
2. In Fig.1, the colony-forming efficiency was very low ($0\sim 0.3\%$) at clonal density ($50\sim 100$ cells/cm²) in the presence of Y-27632. The results appeared to be inconsistent with a large number of previously published studies, and the authors need to check the accuracy of this result.
3. "Except for a slight increase...there was a difference observed between 0.25 μ M and each of the other concentrations: 0.5, 1, 2, 4 μ M (Fig. 1D)", The statement is confusing. Fig. 1C annotated significant differences but the authors claimed no differences, where is 4 μ M in fig.1D.
4. Numerous studies have shown that the long-term treat of Y-27632 lead to spontaneous differentiation of primed hPSCs, and does continuous TTNPB treatment lead to the differentiation of primed hPSCs?
5. The combination of Y-27632 and TTNPB showed higher apoptotic signals in compared to TTNPB alone (Supplementary Fig. 1), this is contradictory to Fig. 1E, F.
6. How is the adhesion ratio of hESCs calculated in Fig. 2A?
7. The combination of Y-27632 and TTNPB showed a lower adhesion ratio and proliferative capacity in comparison to TTNPB alone (Fig. 2A and Fig. 2C), why?
8. "we found better aggregate formation and expansion of both hESCs and hiPSCs at 0.5 μ M compared to other concentrations...", supplementary Fig. 2 did not support the author's statement.
9. "TTNPB-treated cells showed about 80% survival rate 24h after seeding single cells in suspension culture, whereas Y-27632 showed about 60% survival rate for both hESCs and hiPSCs (Fig. 3A, B)", how is the survival rate calculated?
10. "hPSC aggregate morphology differed between Y-27632 and TTNPB treatment as we observed smoothen edges with empty cores in both Y-27632 and Y-27632+TTNPB-treated conditions, whereas TTNPB-treated aggregates showed uneven edges with cell-filled cores (Fig. 3E)", this phenomenon has been reported in a previous study (<https://doi.org/10.1016/j.biomaterials.2020.120015>, Fig. S3A, B), can the authors discuss the similarities or differences between the two?
11. In the brightfield images of Fig. 3E and supplementary Fig. 2A, B, why the differences in the density, size of colonies and their apoptosis are very apparent ?
12. Were the results in Fig. 4 from hESCs or hiPSCs? The statement is not rigorous in section of "Characterization of TTNPB-expanded clonal colonies".
13. Page 8, Line17, the authors mentioned that "Immunostaining of hPSCs also showed higher expression of CLDN-2 and HoxA1 in TTNPB-treated cells compared to Y-27632, why data were not shown?
14. The authors detected the knockout of HoxA1 and CLDN-2 only by PCR, additional sequencing and WB data were needed to determine the knockout of CLDN-2 and HoxA1.
15. Page 27, Line14. "PCR surveyor assay for detecting disrupted HoxA1". HoxA1 should be replaced with CLDN-2.

Reviewer #3 (Remarks to the Author):

This study evaluated the effects of retinoic acid analog on hPSC colony-forming efficiency. hPSCs were seeded at extremely low density (50 – 100 cells/cm²). The authors concluded that TTNPB promotes clonal expansion of hPSCs. This paper will have value to hPSC community. However, some major concerns must be addressed first:

1. In the method section, the source of hESC and iPSC lines must be clearly specified. While H9 is a

well-known hESC line, the 4YF iPSC line should be described or cited if published before.

2. Fig 1 C & D are misleading. Cells were not only treated with TTNPB, but also treated with ROCK inhibitor Y27632 in these two panels. The authors should clearly label Y27632 in the figure.

3. It is needed to perform experiments on clonal colony formation on differing substrates other than Laminin 511, such as Matrigel.

4. Another common method for generating single cell clonal lines is by performing serial dilutions of cell suspension in a 96 well plate. It would be valuable to see if TTNPB treatment increases single cell colony formation using this method.

5. There are many small molecules belonging to retinoic acid analog family. The authors should try other small molecules and test whether they also can promote clonal expansion of hPSCs.

In addition, RAR α agonist (AM580), RAR β agonist (CD 2314), and RAR γ agonist (BMS961) should be used to evaluate which subtype of RAR is needed for activation to promote clonal expansion. This will greatly improve our understanding of the mechanism.

6. RNA-seq data must be submitted to NCBI, so others could assess the data.

For RNA-seq experiment, the untreated control needs to be better defined.

How long were these clonal hPSCs expanded for and for how many passages?

More experimental and control group would be valuable, such as TTNPB vs. Y vs. TTNPB+Y vs. other retinoic acid analog chemicals.

Finally, UMAP analysis and gene ontology between these groups would be helpful to determine transcriptome differences.

7. For HoxA1 and CLDN2 knockout lines, the authors should describe more details about their experiment, such the guide RNA sequences used, how the single cell clonal lines were picked, and CRISPR/Cas9 delivery method.

Also, genomic sequencing at the cutting sites of these clonal lines needs to be performed to assess mono/bi allelic KO, if they are in fact clonal, and the indels that were generated.

Minor issues

Page 4, Line 7: Describe how TTNPB structurally differs from retinoic acid

Page 5, Line 3: Should read "in comparison to the" or "when compared to the"

Page 7, Lines 7 to 10: Discuss this interesting finding more in detail in the discussion section

Reviewers' comments:

We would like to thank all the reviewers for their valuable comments and suggestions. We have responded point by point to their comments and revised the manuscript with yellow highlighted lines.

Reviewer #1 (Remarks to the Author):

In this manuscript, the authors reported that TTNPB, a retinoic acid agonist, promotes colony expansion from dissociated human PSCs and is useful for clone isolation. They identified two factors, CLDN2 and HOXA2, as effector molecules for the TTNPB's action. The authors claimed that CLDN2 regulates cytoskeletal dynamics and YAP activity while HOXA1 upregulated the genes associated with cell-cell adhesion and proliferation. The effect of Y27632+TTNPB seems to be drastic and impressive, but the manuscript includes some concerns that require clarification.

Comment 1. The authors showed that combined treatment of dissociated hPSCs with Y27632+TTNPB showed a greater effect on colony formation than each reagent alone (Figure 1). However, the initial adhesion and cell survival/proliferation are not so different when cells treated with TTNPB only or Y+T were seeded (Figure 2, 3). What is the cause for the difference in colony formation between TTNPB alone and Y+T treatment?

Response 1. We agree with the reviewer that the combined treatment with Y-27632+TTNPB has a greater effect on clonal colony formation compared to TTNPB alone. However, the experimental procedures were different between Fig. 1 and Fig. 2/3. In Fig. 1, we cultured cells in clonal density (50-100 cells/cm²), whereas, in Fig. 2 and 3, cells were seeded in passaging density (1x10⁴ cells/cm²) which is 100 times higher number than the clonal density. The higher ratio of seeding cells gave higher colony formation (Fig. 2, 3), whereas, the lower seeding density had lower colony formation efficiency (Fig. 1). We explained this phenomena in page 10 lines 19-23 and page 11 lines 1-4.

Comment 2. ROCK inhibitors have been reported to block PSC death induced by myosin-dependent cellular contraction. A simple way to show that TTNPB works through different mechanisms from ROCK inhibitors is to determine whether TTNPB affects myosin activity or cellular contraction of dissociated PSCs.

Response 2. We agree with the reviewer on ROCK inhibitors mechanisms of action. We have tested actin-myosin contractility in hESCs by treating the cells with TTNPB and Y-27632 in the current revision. We observed comparatively compact colonies in TTNPB-treated condition, whereas, colonies were loosely connected in Y-27632-treated cells. The immunostaining showed compact F-actin in TTNPB-treated cells, whereas, Y-27632 showed spread out F-actin. We also observed different phenomena in myosin expression in both conditions. Myosin was compact/less expressed in cell cytoplasm of TTNPB-treated condition, whereas it was spread out in Y-27632-treated condition (**Fig. 2F**). Therefore, the mechanism of blocking hESC apoptosis is different in TTNPB-treated condition compared to Y-27632-treated condition in terms of actin-myosin contractility. We have revised results (page 7 lines 2-6), discussion (page 13 line 23, page 14 lines 1-3) and the materials and methods (page 18 lines 19-21) accordingly.

Comment 3. KO experiments seem key for the author's claim about the molecular mechanism of TTPNB action, but the manuscript lacks important information on this part. The detailed descriptions of KO line generation are required (gRNA targets, how many KO clones are isolated and examined? and so on). It is also needed to confirm the expression of HOXA1/CLDN2 is actually defective at protein levels.

Response 3. Thanks for your comments. We have revised the manuscript describing the KO lines generation in detail in the materials and methods section at page 20 lines 18-23 and page 21 lines 1-18.

We did Western blot to detect the disruption of HoxA1/CLDN2 in protein level, but two of our attempts failed because of low quality of antibodies. We tried at least 2 antibodies from two different vendors, but none of them were successful in detecting the HoxA1/CLDN2 protein due to their

impurities. We did DNA-sequencing as the Western blot was not successful to confirm the deletion of both HoxA1 and CLDN2 genes. DNA sequencing is considered one of the major alternatives to Western blot. We found that HoxA1 knockout clones have 8 nucleotides deleted compared to the wild type clone (Fig. 6A, B). CLDN2 knockout clones also showed 4 nucleotides deletion compared to the wild type clones (Fig. 7A, B). These results indicate that both HoxA1 and CLDN2 were disrupted successfully in both KO lines. We described the DNA sequencing data in the manuscript at page 9 lines 9-11 and lines 17-19, and page 21 lines 14-18.

Comment 4. The authors proposed that TTNPB upregulates HOXA1/CLDN2, which in turn promotes the formation of TJ and AJ (shown in Figure 8D). However, no data are showing TJ/AJ formation in TTNPB-treated cells. As they mentioned in Introduction, ROCK inhibitors have been also shown to promote cell-cell adhesion. If the final output caused by these two reagents are similar (promoting/stabilizing cell-cell adhesion?), why TTNPB are more effective than ROCK inhibitors?

Response 4. Figure 8D is a hypothetical assumption which is drawn based on our results and other published papers on the mechanism of HoxA1/CLDN2. However, we observed that TTNPB promotes adherens junction by higher expression of E-cadherin compared to Y-27632 which is described in Fig. 8A. As we found upregulation of CLDN2 expression in TTNPB-treated cells (described in Fig. 5A), we hypothesize that TTNPB promotes the formation of tight junctions and adherens junction in Fig. 8D.

TTNPB is more effective than ROCK inhibitor (Y-27632) as it promotes higher expression of tight and adherens junction proteins (E-cadherin, CLDN2), self-renewal (YAP, Bcl-2 and Cyclin D1) and proliferation (Ki-67) genes (Fig. 5, Fig. 8A) which helps to increase survival and proliferation of dissociated single cells. The expression of these genes with Y-27632 are not as robust as TTNPB which clarifies TTNPB's supremacy in promoting/stabilizing cell-cell adhesion and proliferation. We described these phenomena in the discussion section page 13 lines 8-21.

Comment 5. Suppl Figure 1 is too weak to claim that TTNPB diminished apoptosis of dissociated PSCs. It would be also confusing that green fluorescent signals were detected in the Y+T sample. More quantitative data for cell death/survival are required.

Response 5. Supplementary Fig. 1 is a supporting data of Fig. 2A. As evidenced quantitatively in Fig. 2A, TTNPB showed higher survival ratio compared to both Y-27632 and Y+T conditions with H9 cell line. In supplementary Fig. 1, TTNPB-treated cells showed less green fluorescent signals indicating high survival ratio, whereas Y-27632-treated cells showed higher green fluorescent signals indicating low survival of dissociated cells. Y+T-treated condition also showed higher green fluorescent signals indicating slightly less survival in this condition with H9 cells. We confirmed that Fig. 2A is a quantitative representative of supplementary Fig. 1. We explained these phenomena at page 6 lines 8-17.

Comment 6. Figure legends are not sufficient and reader-friendly. Which timepoint the number of colonies was counted? The concentration of TTNPB used is not specified (except Figure 1C-D). What is 'survival ratio' and how did the authors determine it (Figure 3)? Materials and methods section is also insufficient. The procedures for the quantification of cell death or survival, KO clone generation, DEG identification from RNA-seq data and PCR surveyor assay should be described.

Response 6. Thanks for your comments on these points. We have revised the figure legends providing more details on experimental procedures at pages 28-31. We also revised the materials and methods section by providing detail methodology on cell death/survival, KO clone generation, RNA-seq data and PCR surveyor assay at page 15 lines 19-23, page 16 lines 1-4, page 17 lines 4-7 and lines 16-20, page 20 lines 7-23, page 21 lines 1-18.

Reviewer #2 (Remarks to the Author):

In this study, Nath et al. reported improved clonal expansion and suspension culture of hPSCs using a retinoic acid analogue, TTNPB, which acts, in part, by promoting cellular adhesion and self-renewal through the upregulation of Claudin 2 and HoxA1. I have the following concerns about the data in the manuscript.

Comment 1. How is the colony forming efficiency (Fig. 1A, B) calculated? It is difficult to calculate the colony forming efficiency when passaged at a high cell density ($3\sim 5 \times 10^4$ cells/cm²), I want to know how the colony forming efficiency (over 70%) is calculated in this study.

Response 1. Thanks for your comments. We agree with the reviewer that calculating colony forming efficiency at high density is difficult, especially seeding at ($3\sim 5 \times 10^4$ cells/cm²). We acknowledge that we have mistaken in labelling the figure which should be “cell survival ratio” for the Fig. 1A, B. For calculating the cell survival ratio, we divided the total numbers of cells counted at day 1 of seeding with the total number of seeding cells at day 0. We revised the figure label for Fig. 1A, B and described the method in the materials and methods section at page 16 lines 19-23 and page 17 lines 1-7.

Comment 2. In Fig. 1, the colony-forming efficiency was very low (0~0.3%) at clonal density (50~100 cells/cm²) in the presence of Y-27632. The results appeared to be inconsistent with a large number of previously published studies, and the authors need to check the accuracy of this result.

Response 2. We evaluated TTNPB in two separate seeding densities: (i) at clonal density (50~100 cells/cm², Fig. 1) and (ii) at passaging density which is 100 times higher than clonal density (1×10^4 cells/cm², Fig. 1A/ Fig. 2). In most of the papers published on clonal density, they described 1×10^4 cells/cm² as clonal density which is actually not a clonal density as it is difficult to get single cells with that high number of seeded cells. In our study, we used 50~100 cells/cm² as clonal density which might sound inconsistent with other studies as the clonal seeding densities are different between our study and their studies. We believe that our results of hPSC expansion with the clonal density are accurate.

Comment 3. “Except for a slight increase...there was a difference observed between 0.25 μ M and each of the other concentrations: 0.5, 1, 2, 4 μ M (Fig. 1D)”, The statement is confusing. Fig. 1C annotated significant differences but the authors claimed no differences, where is 4 μ M in Fig. 1D.

Response 3. Thanks for your comment. We revised this statement for Fig. 1C, D in page 5 lines 2-5.

Comment 4. Numerous studies have shown that the long-term treat of Y-27632 lead to spontaneous differentiation of primed hPSCs, and does continuous TTNPB treatment lead to the differentiation of primed hPSCs?

Response 4. Thanks for your comment. In our current study, we mostly focused on short-term expansion of hPSC and their mechanisms of action using TTNPB. We did not consider the long-term effect of TTNPB in this study. We hope to explore this in a future study.

Comment 5. The combination of Y-27632 and TTNPB showed higher apoptotic signals in compared to TTNPB alone (Supplementary Fig. 1), this is contradictory to Fig. 1E, F.

Response 5. Thanks for your comment. There is a significant difference in experimental setup between Fig. 1E, F and Supplementary Fig. 1. Figure 1E, F was performed at clonal density (50~100 cells/cm²), whereas Supplementary Fig. 1 was performed at passaging density (1×10^4 cells/cm²) to support the phenomena explained in Fig. 2A. However, we agree with the reviewer that the combined treatment with Y-27632+TTNPB has a greater effect on clonal colony formation compared to TTNPB alone as shown in Fig. 1E, F. We observed similar phenomena in Supplementary Fig. 1 as well. Although the Y+T condition in Supplementary Fig. 1 showed higher apoptotic signal (green), it also showed higher colony formation and cell survival compared to TTNPB alone (Fig. 2A). We verified it

by quantitatively analyzing the cell expansion in Fig. 2A, where TTNPB alone or Y+T showed a similar survival ratio at passaging density, with no significant difference. We interpreted this in the results section at page 6 lines 8-17.

Comment 6. How is the adhesion ratio of hESCs calculated in Fig. 2A?

Response 6. Cell adhesion was calculated by dividing total cell number on day 1 with the total number of seeded cells on day 0. We described the method in the materials and methods section at page 16 lines 20-23 and page 17 lines 1-7.

Comment 7. The combination of Y-27632 and TTNPB showed a lower adhesion ratio and proliferative capacity in comparison to TTNPB alone (Fig. 2A and Fig. 2C), why?

Response 7. We have explained partially this phenomena in our response in Comment 5. Although the adhesion ratio and final cell numbers in hESC did not show significant difference between TTNPB and Y+T conditions, we assumed that it might be cell line dependent. We did not observe any difference in hiPSC line with the same conditions.

Comment 8. “we found better aggregate formation and expansion of both hESCs and hiPSCs at 0.5 μ M compared to other concentrations...”, supplementary Fig. 2 did not support the author's statement.

Response 8. Thanks for your comments. We revised our statement in the Results section at page 7 lines 9-12.

Comment 9. “TTNPB-treated cells showed about 80% survival rate 24h after seeding single cells in suspension culture, whereas Y-27632 showed about 60% survival rate for both hESCs and hiPSCs (Fig. 3A, B)”, how is the survival rate calculated?

Response 9. Aggregates were dissociated into single cells using TrypLE Express with 0.5mM EDTA and 10 μ M Y-27632 on day 1 to determine the survival ratio. Cells were counted using an automatic cell counter. Cell survival ratio was calculated by dividing total cell number on day 1 with the total number of seeded cells on day 0. We have updated the materials and methods section at page 17 lines 16-20.

Comment 10. “hPSC aggregate morphology differed between Y-27632 and TTNPB treatment as we observed smoothen edges with empty cores in both Y-27632 and Y-27632+TTNPB-treated conditions, whereas TTNPB-treated aggregates showed uneven edges with cell-filled cores (Fig. 3E)”, this phenomenon has been reported in a previous study (<https://doi.org/10.1016/j.biomaterials.2020.120015>, Fig. S3A, B), can the authors discuss the similarities or differences between the two?

Response 10. Thanks for your suggestion. We have referred your recommended paper in our manuscript and discussed the aggregate properties that we observed in our study at page 12 lines 16-23 and page 13 lines 1-5.

Comment 11. In the brightfield images of Fig. 3E and supplementary Fig. 2A, B, why the differences in the density, size of colonies and their apoptosis are very apparent ?

Response 11. The differences in density and size of aggregates are apparently showing different because of the difference in magnification during image capture. Fig. 3E was captured in low magnification and scale bars are 200 μ m, whereas Supplementary Fig. 2A, B were captured in higher magnification and scale bars are 100 μ m. We believe that there is no difference in cell survival and growth in both Fig. 3E and Fig. 2A, B.

Comment 12. Were the results in Fig. 4 from hESCs or hiPSCs? The statement is not rigorous in section of “Characterization of TTNPB-expanded clonal colonies”.

Response 12. We performed similar experiments for both hESCs and iPSCs. The results shown in Fig. 4 are from hESCs, which are indicated in Figure 4 legends. We explained the pluripotency results properly in this revision in the result section at page 8 lines 1-10.

Comment 13. Page 8, Line17, the authors mentioned that “Immunostaining of hPSCs also showed higher expression of CLDN-2 and HoxA1 in TTNPB-treated cells compared to Y-27632, why data were not shown?”

Response 13. We did not show CLDN2 and HoxA1 immunostaining results because they were not as stunning as we expected because of impurity of antibodies. We observed the similar issue with the same antibodies for the Western blot analysis of CLDN2 and HoxA1 analysis. We have revised our statement at page 9 lines 3-5.

Comment 14. The authors detected the knockout of HoxA1 and CLDN-2 only by PCR, additional sequencing and WB data were needed to determine the knockout of CLDN-2 and HoxA1.

Response 14. Thanks for your comments. We did Western blot to detect the disruption of HoxA1/CLDN2 in protein level, but two of our attempts failed because of low quality of antibodies. We tried at least 2 antibodies from two different vendors, but none of them were successful in detecting the HoxA1/CLDN2 protein due to their impurities. We did DNA-sequencing as the Western blot was not successful to confirm the deletion of both HoxA1 and CLDN2 genes. DNA sequencing is considered one of the major alternatives to Western blotting. We found that HoxA1 knockout clones have 8 nucleotides deleted compared to the wild type clone (Fig. 6A, B). CLDN2 knockout clones also showed 4 nucleotides deletion compared to the wild type clones (Fig. 7A, B). These results indicate that both HoxA1 and CLDN2 were disrupted successfully in both KO lines. We described the DNA sequencing data in the manuscript at page 9 lines 9-11 and lines 17-19, and page 21 lines 14-18.

Comment 15. Page 27, Line14. “PCR surveyor assay for detecting disrupted HoxA1”. HoxA1 should be replaced with CLDN-2.

Response 15. Thanks for your comment. We have revised the figure legends by correcting our mistakes at page 31 lines 4.

Reviewer #3 (Remarks to the Author):

This study evaluated the effects of retinoic acid analog on hPSC colony-forming efficiency. hPSCs were seeded at extremely low density (50 – 100 cells/cm²). The authors concluded that TTNPB promotes clonal expansion of hPSCs. This paper will have value to hPSC community. However, some major concerns must be addressed first:

Comment 1. In the method section, the source of hESC and iPSC lines must be clearly specified. While H9 is a well-known hESC line, the 4YF iPSC line should be described or cited if published before.

Response 1. Thank you for your comments. We have revised our manuscript by providing the sources of hESC and iPSC lines at page 15 lines 3-4.

Comment 2. Fig 1 C & D are misleading. Cells were not only treated with TTNPB, but also treated with ROCK inhibitor Y27632 in these two panels. The authors should clearly label Y27632 in the figure.

Response 2. Fig. 1C & D are for screening of TTNPB concentrations for the expansion of hPSCs. Therefore, all cells in these conditions are treated only with TTNPB in the Fig. 1C & D.

Comment 3. It is needed to perform experiments on clonal colony formation on differing substrates other than Laminin 511, such as Matrigel.

Response 3. Thanks for your suggestion. We have performed experiments on clonal colony formation on both Laminin and Matrigel (Supplementary Fig. 1). We found that Laminin is superior to Matrigel in clonal colony formation while cells treated with TTNPB and Y-27632. We described it in our manuscript at page 6 lines 1-5.

Comment 4. Another common method for generating single cell clonal lines is by performing serial dilutions of cell suspension in a 96 well plate. It would be valuable to see if TTNPB treatment increases single cell colony formation using this method.

Response 4. Thanks for your suggestion. We agree that serial dilution in a 96-well plate is an alternative method for generating clonal colonies from single cells. As our current method is validated and we got satisfactory results from it, we would like to consider your suggestion for our future study.

Comment 5. There are many small molecules belonging to retinoic acid analog family. The authors should try other small molecules and test whether they also can promote clonal expansion of hPSCs.

In addition, RAR α agonist (AM580), RAR β agonist (CD 2314), and RAR γ agonist (BMS961) should be used to evaluate which subtype of RAR is needed for activation to promote clonal expansion. This will greatly improve our understanding of the mechanism.

Response 5. Thanks for your valuable suggestions. We came to investigate TTNPB's effect on cell survival based on our previous study in cellular reprogramming where we tried TTNPB as a chemical reprogramming agent for reprogramming human skin fibroblast to hiPSC. As apoptosis is a critical factor in lowering reprogramming efficiency, we also screened other small molecules for preventing apoptosis. For example, Pioglitazone, Rosiglitazone, C3 exoenzyme, and HA-1077 etc., but none of them were as efficient as TTNPB. We will keep exploring other small molecules for our future study.

Inhibition of RAR is also an interesting idea to check TTNPB's effect on promoting clonal expansion. However, TTNPB mostly worked on upregulating CLDN2 and HoxA1 genes (based on RNA-seq data), which were totally different genes than the RARs. For that reason, we were not interested in

investigating the RARs as it is not the dominant pathway for clonal expansion of hiPSCs. We will keep your suggestion in mind for our future study.

Comment 6. RNA-seq data must be submitted to NCBI, so others could assess the data.

For RNA-seq experiment, the untreated control needs to be better defined.

How long were these clonal hPSCs expanded for and for how many passages?

More experimental and control group would be valuable, such as TTNPB vs. Y vs. TTNPB+Y vs. other retinoic acid analog chemicals.

Finally, UMAP analysis and gene ontology between these groups would be helpful to determine transcriptome differences.

Response 6. We have submitted all raw data from RNA-seq to the NCBI-GEO. All data can be found in the following accession number in the GEO BioProject # SUB11997390. We have updated it in the manuscript at page 22 lines 16-18.

The untreated controls were described in the manuscript at page 20 lines 8-16.

The clonal expansion of hPSCs were performed for 7 days. We have mentioned the culture conditions in the materials and methods section at page 15 lines 19-23 and page 16 lines 1-4. However, we did not expand them for long-term culture as it was not a scope for this manuscript. We may investigate the long-term culture using TTNPB in our future study.

Thanks for suggesting more experimental control groups. In our current study, we focused on only TTNPB and its mechanism on promoting clonal expansion using two hPSC lines. Therefore, we were not focused in investigating any other combination.

We believe that we provided most of the RNA-seq analysis data to support our claims in this manuscript.

Comment 7. For HoxA1 and CLDN2 knockout lines, the authors should describe more details about their experiment, such the guide RNA sequences used, how the single cell clonal lines were picked, and CRISPR/Cas9 delivery method.

Also, genomic sequencing at the cutting sites of these clonal lines needs to be performed to assess mono/bi allelic KO, if they are in fact clonal, and the indels that were generated.

Response 7. Thanks for your suggestions. We have provided detailed methods for HoxA1 and CLDN2 knockout clones generation including gRNA seq design, CRISPR/Cas9 delivery, and indels generated in the manuscript at page 20 lines 8-23 and page 21 lines 1-18.

The genomic sequences in the knockout clones were analyzed by DNA sequencing. We provided details of DNA sequencing in the manuscript at page 21 lines 14-18, page 9 lines 9-11 and 17-19.

Minor issues

Comment 8. Page 4, Line 7: Describe how TTNPB structurally differs from retinoic acid

Response 8. Thanks for the comment. We revised the manuscript by adding details on TTNPB and retinoid at page 4 lines 6-9.

Comment 9. Page 5, Line 3: Should read “in comparison to the” or “when compared to the”

Response 9. We revised the manuscript at page 6 line 11-12 considering the reviewers comment.

Comment 10. Page 7, Lines 7 to 10: Discuss this interesting finding more in detail in the discussion section

Response 10. Thanks for the comment. We revised the manuscript by discussing more at page 12 lines 16-23 and page 13 lines 1-5.

Reviewers' comments:

Reviewer #1 (Remarks to the Author):

The revised manuscript has been improved, but I point out that there are still concerns that need to be refined before publication, which are described below.

The author responded to my comment 6, but it is surprising that the same calculation value was used to verify the distinct cellular behaviors, cell survival and adhesion (p17, lines 5-6). The authors claim in the abstract that "TTNBP acts, in part, by promoting cellular adhesion", but they provide no direct evidence for cell adhesion. Instead, they quantified cell adhesion by calculating the "Adhesion ratio" in Fig 2A and B, which is by definition the same as the "Survival ratio". If the authors use the same calculation values to validate the cell's survival and adhesion, it is not possible to discuss whether TTNBP affects cell viability or adhesion. If authors conclude that TTNBP promotes cellular adhesion, they should present quantification data for cell adhesion rather than simply counting cell numbers. Similarly, if authors conclude that "TTNBP prevent apoptosis (p14, line 19), they should utilize the standard methods for quantifying apoptosis (e.g, annexin V staining). It might be not generally acceptable to validate and discuss cell survival and adhesion using the same approach of "by dividing total cell number on day 1 with the total number of seeded cells on day 0", since the number of cells at specific timepoints may reflect several effects on different cell activities, survival/death, adhesion to culture surface, proliferation, etc.

Another ambiguity in this study may be the difference in the effects of ROCK inhibitor and TTNBP on cell behaviors. The authors stated in the discussion part (p14, lines 2-3) that "Although Y-27632 prevents PSC apoptosis by myosin-dependent cellular contraction, TTNBP works differently by dampening the myosin activity", but this statement makes it difficult to catch the author's claim. In the author's conclusion, does TTNBP regulate myosin activity or not?; do both ROCK inhibitor and TTNBP prevent apoptosis, but by different mechanisms? It should be clarified what is different and what is similar about the actions of ROCK inhibitor and TTNBP?

minor comments

- Which antibody was used for myosin staining? heavy chains or light chains? I cannot find what reagents were used to stain for F-actin? (phalloidin? antibody? or others?)
- In the revised manuscript, the authors mention that two KO clones were examined. It should be stated whether similar results were obtained from both clones or not.
- Should describe how to determine DEGs in detail. What type of significant test was used?
- A detailed explanation for qPCR experiments is still lacking. What is the "relative gene expression fold" in Fig.8 A-C ? What is set to "1"?

Reviewer #2 (Remarks to the Author):

All the raised concerns were adequately addressed.

Reviewer #3 (Remarks to the Author):

The authors refused to do more experiments to address some of my major concerns, such as generating single cell clonal lines is by performing serial dilutions of cell suspension in a 96 well plate; and testing several RAR small molecules to understand the mechanism of the finding.

This is unresponsive to reviewers' valuable suggestions.

These two experiments should be done and can provide very valuable information and enhancing our understanding about single cell survival of stem cells.

Reviewers' comments:

We would like to thank all the reviewers for their valuable comments and suggestions. We have responded point by point to their comments and revised the manuscript with yellow highlighted lines.

Reviewer #1 (Remarks to the Author):

The revised manuscript has been improved, but I point out that there are still concerns that need to be refined before publication, which are described below.

Comment: The author responded to my comment 6, but it is surprising that the same calculation value was used to verify the distinct cellular behaviors, cell survival and adhesion (p17, lines 5-6). The authors claim in the abstract that “TTNBP acts, in part, by promoting cellular adhesion”, but they provide no direct evidence for cell adhesion. Instead, they quantified cell adhesion by calculating the “Adhesion ratio” in Fig 2A and B, which is by definition the same as the “Survival ratio”. If the authors use the same calculation values to validate the cell's survival and adhesion, it is not possible to discuss whether TTBBP affects cell viability or adhesion. If authors conclude that TTNPB promotes cellular adhesion, they should present quantification data for cell adhesion rather than simply counting cell numbers. Similarly, if authors conclude that "TTNPB prevent apoptosis (p14, line 19), they should utilize the standard methods for quantifying apoptosis (e.g, annexin V staining). It might be not generally acceptable to validate and discuss cell survival and adhesion using the same approach of "by dividing total cell number on day 1 with the total number of seeded cells on day 0", since the number of cells at specific timepoints may reflect several effects on different cell activities, survival/death, adhesion to culture surface, proliferation, etc.

Response: Thank you for your constructive comments. We agree with the reviewer and cleared our claim for both cell survival and cell adhesion after TTNPB treatment in this revision. We performed additional experiment for supporting our hypothesis that TTNPB increases hPSC survival by measuring caspase-3 (a key marker in apoptosis pathway) expression with the Western blot technique (Supplementary Fig. 2B, C). We also performed an additional experiment for measuring cell adhesion by crystal violet cell adhesion assay in this revision (Supplementary

Fig. 2D, E). We revised the manuscript after considering the data from the new experiments on page 6 lines 9-23, page 7 lines 1-3, page 22 lines 19-23, and page 23 lines 1-21.

Comment: Another ambiguity in this study may be the difference in the effects of ROCK inhibitor and TTNPB on cell behaviors. The authors stated in the discussion part (p14, lines 2-3) that “Although Y-27632 prevents PSC apoptosis by myosin-dependent cellular contraction, TTNPB works differently by dampening the myosin activity”, but this statement makes it difficult to catch the author’s claim. In the author's conclusion, does TTNPB regulate myosin activity or not?; do both ROCK inhibitor and TTNPB prevent apoptosis, but by different mechanisms? It should be clarified what is different and what is similar about the actions of ROCK inhibitor and TTNPB?

Response: We thank the reviewer for the constructive comment. We revised the manuscript to clarify the mechanisms of both Y-27632 and TTNPB in terms of their similarity and differences on page 14 lines 8-19.

Minor comments

- Which antibody was used for myosin staining? heavy chains or light chains? I cannot find what reagents were used to stain for F-actin? (phalloidin? antibody? or others?)

Response: We thank the reviewer for the comment. We used the recombinant anti-non-muscle Myosin IIA antibody (EPR8965, ab138498), which is mentioned in revised manuscript on page 19 line 10. We could not find whether the Myosin IIA antibody is heavy chain or light chain as the vendor did not provide the info. We used a Phalloidin antibody (Invitrogen, cat# R415), which is conjugated to the red-orange fluorescent dye for staining F-actin. We included the F-actin details in the manuscript on page 19 line 9.

- In the revised manuscript, the authors mention that two KO clones were examined. It should be stated whether similar results were obtained from both clones or not.

Response: Thanks for the comment. We confirmed both KO clones for successful deletion of respective genes and showed the result from a representative clone. We revised the manuscript by stating the confirmation of gene deletion of both KO clones at page 9 line 17-18 and page 10 line 4-5.

- Should describe how to determine DEGs in detail. What type of significant test was used?

Response: Thanks for the comment. We did not understand what did the reviewer mean by DEGs. We used Student's T test for evaluating the significance from three replicate experiments. We revised the manuscript describing the statistical analysis at page 25 lines 4-8.

- A detailed explanation for qPCR experiments is still lacking. What is the "relative gene expression fold" in Fig.8 A-C? What is set to "1"?

Response: Thanks for the comment. The relative gene expression fold was determined by normalizing the data with the GAPDH expression. We revised the figure legend of Fig. 8 A-C. We added q-PCR and RT-PCR protocol in the revised manuscript at page 23 lines 23, page 24 lines 1-23, page 25 line 1-2.

Reviewer #2 (Remarks to the Author):

All the raised concerns were adequately addressed.

Response: Thank you for your consideration.

Reviewer #3 (Remarks to the Author):

The authors refused to do more experiments to address some of my major concerns, such as generating single cell clonal lines is by performing serial dilutions of cell suspension in a 96 well plate; and testing several RAR small molecules to understand the mechanism of the finding.

This is unresponsive to reviewers' valuable suggestions.

These two experiments should be done and can provide very valuable information and enhancing our understanding about single cell survival of stem cells.

Response: We thank the reviewer for the comments. We definitely respect the reviewer's comments which make our manuscript better. In the last revision, the reviewer raised a total of 10 concerns, and we replied to them humbly. However, we believe the currently suggested two experiments are either out of scope or they do not add much value to our current study. Moreover, performing a serial dilution experiment is repetitive to our current cell tracing experiment described in Fig. 1 I and J. Testing several RAR small molecules is a separate study, which we may consider for our future study.

REVIEWERS' COMMENTS:

Reviewer #1 (Remarks to the Author):

I think the revised manuscript has been improved substantially. I support publication in Communications Biology.